

# On the representation of aerosol activation and its influence on model-derived estimates of the aerosol indirect effect

Daniel Rothenberg[1], Alexander Avramov[2], and Chien Wang[1]

[1]Center for Global Change Science, Massachusetts Institute of Technology, Cambridge, MA
[2]Department of Environmental Sciences, Emory University, Atlanta, GA

*Correspondence to:* Daniel Rothenberg (darothen@mit.edu)

**Abstract.** Interactions between aerosol particles and clouds contribute a great deal of uncertainty to the scientific community's understanding of anthropogenic climate forcing. Aerosol particles serve as the nucleation sites for cloud droplets, establishing a direct linkage between anthropogenic particulate emissions and clouds in the climate system. To resolve this linkage, the community has developed parameterizations of aerosol activation which can be used in global climate models to interactively predict cloud droplet number concentrations (CDNC). However, different activation schemes can exhibit different sensitivities to aerosol perturbations in different meteorological or pollution regimes. To assess the impact these different sensitivities have on climate forcing, we have coupled three different core activation schemes and variants with the CESM-MARC. Although the model produces a reasonable present day CDNC climatology when compared with observations regardless of the scheme used, $\Delta$CDNC between the present and pre-industrial era regionally increase by over 100% in zonal mean when using the most sensitive parameterization. These differences in activation sensitivity lead to a spread of over $0.8\,\mathrm{Wm}^{-2}$ in global average shortwave indirect effect (AIE) diagnosed from the model, a range which is as large as the inter-model spread from the Aero-Com inter-comparison. Model-derived AIE strongly scales with the simulated pre-industrial CDNC burden, and those models with the greatest pre-industrial CDNC tend to have the smallest AIE, regardless of their $\Delta$CDNC. This suggests that present day evaluations of aerosol-climate models may not provide useful constraints on the magnitude of AIE, which will arise from differences in model estimates of the pre-industrial aerosol and cloud climatology.

## 1 Introduction

Interactions between aerosol and water in different phases contribute significant uncertainty towards the assessment of anthropogenic climate change. Much of this uncertainty arises from the role of aerosol particles as nuclei which seed the formation of clouds. Changes in the ambient particle burden influence the microstructure of clouds and their optical properties, leading to an "indirect effect" (AIE) on climate (Twomey, 1977; Albrecht, 1989). Constraining the magnitude of this influence, though, is difficult, and large uncertainties persist despite rapid developments in both modeling and observations (Boucher et al., 2013).

The difficulty in constraining the indirect effect's magnitude on contemporary climate change arises from two different but complementary sources. Ghan et al. (2013) and Carslaw et al. (2013) illustrated using two distinct approaches that lack of constraints on natural aerosol emissions and their pre-industrial size distributions and chemistry contributes a component



of epistemic uncertainty to the problem. However, interactions between aerosols and clouds are also state-dependent; the sensitivity of cloud processes and properties such as precipitation or albedo to aerosol perturbations can vary widely across aerosol and cloud regimes (Quaas et al., 2009; Zhang et al., 2016; McCoy et al., 2017). Thus, efforts to either improve historical constraints on the ambient aerosol burden or improve the simulation of aerosol-cloud interactions could help reduce uncertainty

in AIE.

Details concerning the representation or parameterization of aerosol and cloud processes in global climate models can also influence the magnitude of their simulated AIE. Hoose et al. (2009) showed that in one model, the indirect effect scaled nearly linearly with an artificial constraint placed on the minimum permissible cloud droplet number concentration (CDNC), a tuning parameter first developed to mitigate unrealistically-low droplet numbers in remote maritime clouds. A similar threshold used

to tune droplet autoconversion parameterizations also yielded scaled estimates of AIE in a different model (Golaz et al., 2011). Assumed empirical relationships between ambient sulfate burden and CDNC were shown to contribute largely to the spread in shortwave cloud forcing observed in a previous generation of global climate models (Storelvmo et al., 2009). Furthermore, the fundamental representation of aerosol particle size distribution has also been implicated as a large contributor to a given model's AIE (Kodros and Pierce, 2017).

Here, we consider a fundamental component of aerosol-cloud interactions (ACI hereafter) included in contemporary global climate models—the nucleation of cloud droplets from the ambient aerosol population (also known as "aerosol activation"). Droplet nucleation plays a key role in setting the climatology of CDNC simulated within global models by providing the initial inputs to cloud microphysical processes. In this manner, activation schemes provide a direct linkage between otherwise-independently modeled aerosol and microphysical processes, enabling an explicit representation of the indirect effect. This

explicit representation has been implicated as a critical component necessary to resolve regional aerosol impacts on both warming trends over the 20th century (Ekman, 2014) and changes in precipitation patterns (Wang et al., 2015).

Several activation schemes have previously been developed for use in global climate models (e.g. Abdul-Razzak and Ghan, 2000; Nenes and Seinfeld, 2003; Ming et al., 2006; Shipway and Abel, 2010). However, the subjective choice of activation scheme used in a model can influence its simulated CDNC and ACI. Ghan et al. (2011) found a 10% difference in AIE when

using two different activation schemes in the same global model, despite a 20-50% difference in simulated CDNC, which is much smaller than the typical inter-model spread in AIE (e.g. Lohmann and Ferrachat, 2010; Boucher et al., 2013). Using a similar global model but with more complex aerosol chemistry module, Gantt et al. (2014) dramatically increased the difference in simulated CDNC to 155% for the same two schemes, which led to a change in present day shortwave cloud forcing of 13%. In another study using the same global model, Gettelman (2015) observed a 28% decrease in the indirect effect when altering

the numerics of activation such that droplets are nucleated before other microphysical tendencies are computed.

This work extends these previous literature efforts by quantifying the influence of the representation of activation on estimates of the indirect effect using a suite of state-of-the-science parameterizations coupled to an aerosol-climate model. We include in our suite of parameterizations a sophisticated emulator of droplet nucleation based on an adiabatic parcel model (Rothenberg and Wang, 2017). Furthermore, as a reference, we compare our results to an intercomparison of indirect effect

calculations using a suite of global climate models with different aerosol and microphysical schemes. Although we cannot fully



explore the aerosol and cloud microphysical parameter space over which AIE could be sensitive, these comparisons highlight the importance of the aerosol-cloud coupling in influencing the problem.

This manuscript is organized as follows. Section 2 introduces the MARC aerosol-climate model used to investigate the influence of droplet activation on the indirect effect, as well as the observational and model intercomparison datasets used in this study. In Sects. 3.1.1-3.1.2 we consider how different activation schemes influence the simulation of clouds and radiation in a present-day emission scenario, and their sensitivity to aerosol perturbations. We follow this with analysis in Sects. 3.3-3.5 on the influence of droplet activation on the indirect effect. Section 4 summarizes our findings and discusses implications for future studies. Finally, we include a Supplement documenting MARC's simulation of aerosol physical and radiative forcing climatologies with available observational data.

## 2   Methods

### 2.1   MARC Global Aerosol-Climate Model

In order to assess aerosol impacts on climate, we have utilized the two-Moment, Multi-Modal, Mixing-state-resolving Aerosol model for Research of Climate (MARC) coupled with the National Center for Atmospheric Research's (NCAR) Community Earth System Model (CESM; version 1.2) with the Community Atmosphere Model (CAM; version 5.3). In this CESM-MARC model (MARC hereafter) we replace the default aerosol scheme (the Modal Aerosol Model Liu et al., 2012) with an aerosol physics and chemistry model based on the scheme by Wilson et al. (2001). MARC has previously been used to resolve aerosol physics in both cloud-resolving (Ekman et al., 2004, 2006; Wang, 2005a, b; Ekman et al., 2007; Engström et al., 2008) and climate (Kim et al., 2008; Ekman et al., 2012; Kim et al., 2014) simulations.

MARC explicitly simulates the evolution of a complex mixture of aerosol species, each with an associated lognormal size distribution. Within MARC, the aerosol species are divided into a set of externally-mixed modes, including three distinct sulfate modes (nucleation or "NUC", Aitken or "AIT", and accumulation "ACC"), pure black carbon (BC), and organic carbon (OC). Additionally, MARC resolves two internally-mixed modes, consisting of sulfate-black carbon (MBS) and sulfate-organic carbon (MOS). With the MBS mixture, particles are assumed to consist of a black carbon core coated with a sulfate shell; within the MOS mixture, particles are totally internally mixed according to the volumetric ratio of sulfate and organic carbon present. For each mode, MARC tracks the evolution of total number and mass concentrations. Additionally, MARC tracks the partitioning between carbon and sulfate for both the MOS and MBS modes.

Sulfate particles are formed in MARC via binary nucleation of $H_2SO_4H_2O$ (Vehkamäki, 2002), with prognostic gaseous sulfuric acid predicted by the default CAM interactive sulfur chemistry module (Barth et al., 2000). Both gas-phase oxidation of $SO_2$ and dimethyl sulfide (DMS) provide sources for $H_2SO_4$, as well as aqueous reactions of S(IV) with both $H_2O_2$ and $O_3$. Coagulation between modes produces both pure (externally-mixed sulfate) and mixed (MOS and MBS) particles; the pure carbon (BC and OC) modes age into their mixed counterparts through a prescribed constant-time aging scheme (40 and 20 hours, respectively) which is limited by the availability of $H_2SO_4$ for condensation. Both primary and secondary organic carbon aerosol are emitted into the OC mode; biogenic volatile organic vapors (specifically isoprene and monoterpenes) are





converted upon emission into OC using a simple yield coefficient suggested by Griffin et al. (1999). We assume that both pure carbonaceous modes are hydrophobic.

Dust and sea salt are computed in MARC using a sectional, single-moment (fixed-size) scheme (with mean size bins of $0.16\,\mu\text{m}$, $0.406\,\mu\text{m}$, $0.867\,\mu\text{m}$, $1.656\,\mu\text{m}$ bins for dust and $0.2\,\mu\text{m}$, $2\,\mu\text{m}$, $5\,\mu\text{m}$, $15\,\mu\text{m}$ for sea salt). Sea salt is assumed to be

composed of NaCl, while dust is assumed to be a mixture of minerals (Albani et al., 2014; Scanza et al., 2014). Prescribed size distribution and hygroscopicity parameters for each mode are summarized in Table 1.

The aerosols simulated by MARC fully couple and interact with both the CESM radiative transfer model and its cloud microphysics scheme (through droplet nucleation). Particles from all modes can be lost through dry deposition, gravitational settling and impaction scavenging via precipitation, and each mode undergoes these processes with different efficiencies related to their

size and hygroscopicity (Petters and Kreidenweis, 2007). Additionally, nucleation scavenging occurs in both deep convective and stratiform clouds. In deep convection, prescribed cloud-base supersaturation of 0.1% is assumed to estimate scavenging. However, in stratiform clouds, nucleation scavenging is calculated through a prognostic aerosol activation scheme, taking into account both local meteorology (sub-grid scale updraft speeds) and the total availability of ambient aerosol. Although several aerosol species (sulfate and dust) play a role in heterogeneous ice formation in MARC (following Liu et al., 2007), this process

does not remove ambient aerosol.

MARC adopts the stratiform cloud microphysics scheme from CAM5.3 (Morrison et al., 2008) and includes the updates to code structure and droplet nucleation tendencies referred to as MG1.5 by Gettelman and Morrison (2015). The contribution of droplet nucleation to the cloud droplet number tendency, $\frac{\partial N_d}{\partial t}$, is computed following Ovtchinnikov and Ghan (2005), and can be non-zero in both newly formed and pre-existing clouds. Droplet nucleation is restricted to occur at cloud-base in pre-existing

clouds but can occur at all levels of newly-formed clouds where cloud water mass is predicted to develop. Additionally, $\frac{\partial N_d}{\partial t}$ includes sink terms such as accretion of cloud water, self-collection of hydrometeors, evaporation, autoconversion, advection, and inter-type scavenging of hydrometeors. Autoconversion is parameterized as a function of cloud water content and droplet number, $N_d$ (Khairoutdinov and Kogan, 2000). Ice and mixed-phase cloud microphysics are based on Liu et al. (2007) and Gettelman et al. (2010).

## 25   2.2   Simulation Design and Analysis

We perform a set of simulations with MARC using different activation schemes (see Appendix A for more details). For both the pseudo-analytical *ARG* (Abdul-Razzak and Ghan, 2000) and iterative *nenes* (Morales Betancourt and Nenes, 2014b) schemes, we include both a **_comp** and a **_min_smax** variant, where the latter incorporates a minimum-$S_{\text{max}}$ heuristic and neglects intermodal competition for water vapor during activation. For the *PCM* schemes, we use both the **main4** and **gCCN3** schemes of

Rothenberg and Wang (2017), and the 4th-order OLS scheme of Rothenberg and Wang (2016) for the supplemental heuristic. We also note that for the *nenes* scheme, we apply the kappa-Köhler theory formulation to handle dust instead of the adsorption mechanics implemented by Kumar et al. (2009). Other than the change in activation schemes, the simulations use the same emissions scenarios and physics schemes.





For each scheme, we performed a pair of 6-year simulations using a horizontal grid resolution of $1.9 \times 2.5$ and 30 vertical levels. Each simulation is run with prescribed sea surface temperatures and ice cover running an annual cycle for the year 2000. To focus this work on the indirect effect, we diagnose the aerosol direct radiative effect through additional radiative transfer calls during model run-time, but we do not include this effect in the heating rates used to forward-integrate the model. The

pairs of simulations differ only in their prescribed aerosol and precursor gas emissions; here, we use a present day ("PD") and pre-industrial ("PI") corresponding to the years 2005 and 1850, respectively. Following Kim et al. (2008), we use constant emissions derived using an offline modeling process (Mayer et al., 2000; Wang, 2004) for BC and primary OC; emissions of DMS and VOC (isoprene and monoterpene) vary on a monthly basis. SO2 emissions are taken from the default CESM inventory. Dust emissions are based on modeled wind speeds and land-surface usage and are tuned following Albani et al.

(2014). Similarly, sea salt emissions are dependent on both wind speeds near the surface as well as sea surface temperature and use the original scheme used in CESM (Liu et al., 2012).

For all simulations, we output monthly mean fields and analyze the final 3 years of output for both the PD and PI cases. The change in shortwave cloud radiative forcing between the two cases is diagnosed using a decomposition which takes into account impacts due to surface albedo change (Ghan, 2013):

$$\Delta C_{\text{clean}} = \Delta(F_{\text{clean}} - F_{\text{clear,clean}})$$

where $\Delta$ indicates the difference between the PD and PI simulations, $F_{\text{clean}}$ is the radiative flux calculated neglecting the scattering and absorption of all aerosol, and $F_{\text{clear,clean}}$ further excludes clouds.

Additionally, we output a suite of instantaneous cloud micro- and macrophysical variables sampled at either cloud-top or one kilometer above the surface and saved every three hours over the duration of the simulations. For consistency with the

radiative transfer calculations in the model, the maximum-random overlap hypothesis is used to derive cloud-top quantities. To estimate the sensitivity of the indirect effect and cloud microphysical properties to cloud and aerosol perturbations, we analyze timeseries of the quantities of interest in each grid cell, considering only those where liquid water clouds are present (temperature <-5 C between -60°S and 60°N. From these masked timeseries in each grid cell we compute climatologies of aerosol and cloud radiative microphysical properties which we then compare against to assess sensitivity of the shortwave

cloud radiative effect to aerosol and cloud microphysical perturbation.

### 2.3 Other Data

#### 2.3.1 Satellite Observations

To assess MARC's performance in simulating present-day cloud and radiation fields, we use a climatology of observations derived from satellite-based sensors. Cloud micro- and macro-physical fields were derived from the MODerate Resolution

Imaging Spectroradiometer (MODIS; Collection 5.1). Cloud droplet number is derived from Level 1 data from the same instrument using a technique employing an adiabatic cloud assumption (Bennartz and Rausch, 2017); for this reason, it is only





suitable for maritime cloud regimes equator-ward of 60°. The global radiative budget at the top of the atmosphere is estimated using the climatology from the Clouds and Earth's Radiant Energy System (CERES) Energy Balanced and Filled (EBAF) dataset (Loeb et al., 2009). All data is re-gridded to the MARC simulation grid before time-averaging for analysis.

### 2.3.2  CDNC Observations

Because the MODIS-derived cloud droplet number concentration retrievals have a high degree of uncertainty, we also evaluate simulated droplet numbers against a large collection of *in situ* observations previously compiled by Karydis et al. (2011) (see their Table 2). We compare these observations to instantaneous output of in-cloud droplet number from our present-day simulations, first interpolated to 850 mb, and then averaged over the indicated seasons and observation areas for the final three years in each model run. For observations from a specific location, we locate the model grid-cell containing that location for analysis. As a reference, we include the modeled CDNC corresponding for each observation produced by the chemical transport model simulations performed by Karydis et al. (2011). Figure 1 plots the global distribution of where the observations are sourced.

### 2.3.3  AeroCom Model Comparison

We supplement our simulations by further analyzing an additional set of climate model output from the Aerosol Comparisons between Observations and Models (AeroCom) Indirect Effects Experiment. This intercomparison includes 5 independent aerosol-climate models (CAM5, ECHAM6-HAM2, ModelE-TOMAS, SPRINTARS, and HadGEM3-UKCA), as well as several variations on the core models adjusting the cloud microphysical scheme (CAM5-MG2), the turbuluence closure (CAM5-CLUBB), and the autoconversion scheme (SPRINTARS-KK). Similar to the experiment conducted here, pairs of integrations (using present-day and pre-industrial emissions scenarios) were performed with each model, using the same IPCC emissions scenarios for primary aerosol and precursors (Lamarque et al., 2011). Each simulation uses prescribed sea surface temperatures, sea-ice extent, and atmospheric greenhouse gas concentrations, and was run for a length of five model years. A detailed summary of each model and its suite of parameterizations relevant for modeling the indirect effect can be found in Appendix A of (Zhang et al., 2016).

## 3  Results

### 3.1  Influence of activation scheme versus observations

#### 3.1.1  Cloud droplet number concentration

Predicted cloud droplet number concentration (CDNC) from each variant of MARC are compared against observations sourced from around the globe (Fig. 1) in Fig. 2. MARC generally under-predicts CDNC in regimes where observed CDNC are very high, particularly over polluted continental regions. Both of the *nenes* schemes and each of the *min_smax* schemes (relative to their full-competition reference) are able to simulate the high CDNC values in these regions. In clean marine regimes,



all of the MARC simulations produce too-little variance in simulated CDNC, although most of the comparisons are within $\pm 50\%$ of the observations, which tend to be small. The detailed aerosol, chemistry, and activation treatment in the NASA GMI model utilized by Karydis et al. (2011) produces much better agreement with observations, although their model also tends to consistently predict too much CDNC over continental regions; over half of their reported values are greater than their

corresponding observed CDNC values by $\pm 50\%$.

Distributions of relative error in model-simulated CDNC versus observations aggregated by region are shown in Fig. 3. On average, MARC performs the worst in continental regimes, regardless of activation scheme. In contrast, the average simulated CDNC in clean marine regimes is well-calibrated, but has much higher variance. Polluted maritime regimes tend to have the least variance, and the model performs better in these regimes than over continents. In the global average, though, CDNC

burden is relatively well-predicted in comparison with recent modeling estimates. Estimates of global-average CDNC in the simulations performed here range from 60-91 $\mathrm{cm}^{-3}$ (for the *arg_comp* and *nenes_min_smax* cases, respectively). This is mostly in agreement with recent studies, albeit on the lower side of estimates (75-135 $\mathrm{cm}^{-3}$ by Penner et al. (2006) using a suite of models employing the Abdul-Razzak and Ghan (2000) activation scheme; 83 $\mathrm{cm}^{-3}$ by Leibensperger et al. (2011) using an empirical relationship between aerosol and droplet number; 96 $\mathrm{cm}^{-3}$ by Barahona et al. (2011) using an earlier variant of the

Morales Betancourt and Nenes (2014b) activation scheme).

For a more rigorous assessment of simulated CDNC, we compare MARC fields to CDNC derived from MODIS observations (Bennartz and Rausch, 2017) in Fig. 4. Enhancement in CDNC downwind and in the vicinity of continents and anthropogenic emissions sources are clearly visible in the satellite dataset, particularly in the regions offshore of the United States and China. Averaged over the entire oceanic region under consideration, MARC under-predicts CDNC by 45-56% depending on which

activation scheme is used. However, CDNC is consistently too small in several regions regardless of activation scheme, particularly in both the north and south Atlantic, in the portion of the Southern Ocean that lies south of the Indian Ocean, and in the north Pacific. The only oceanic region where the model over-estimates CDNC is in the equatorial upwelling region of the eastern Pacific. Enhancement of CDNC by anthropogenic aerosol in coastal regions is best-captured by the *nenes* and *PCM* schemes.

Although using different activation schemes does not directly perturb the simulated aerosol distributions in MARC, the two-way coupling facilitated by nucleation scavenging can indirectly influence average aerosol number concentrations. In these simulations, the PD accumulation mode number concentration over the oceans is 31-40% smaller in the simulations using the *nenes* and *PCM* activation schemes versus the *ARG*. This is likely because the former two schemes tend to nucleate more droplets, given a similar aerosol population. The attendant increase in nucleation scavenging decreases accumulation mode

number, which then depresses potential cloud droplet number. As a result, the difference in the long-term average CDNC between the different schemes is not as large as it otherwise might be, hence the similar distributions of error relative to MODIS-derived CDNC. The region of too-high CDNC simulated by MARC in the eastern equatorial Pacific coincides with a region of enhanced, persistent deep convection and precipitation in the model.





### 3.1.2 Clouds and radiation

Compared to the original version of CESM/CAM5.3, the inclusion of an alternative aerosol formulation does not substantially change the model's simulated cloud and radiation fields, as illustrated in Fig. 5. To demonstrate the extent to which altering the activation scheme can influence these fields on the large-scale, we have included in Fig. 5 zonal averages computed from an

ensemble of aerosol-climate models, four of which are themselves variants on the CAM5.3 with alternative microphysics (Gettelman and Morrison, 2015) and/or moist turbulence scheme (Bogenschutz et al., 2013). Except in the Northern Hemisphere sub-tropics, MARC tends to under-predict total aerosol optical depth (AOD) relative to both observations and the reference models (Fig. 5a). This is generally the case for all of the reference models as well, and MODIS estimates of AOD are thought to be biased high over oceanic regimes (Levy et al., 2013). However, MARC consistently predicts smaller AOD than the other

models considered here. As previously noted in Sect. 3.1.1, there are differences in simulated accumulation-mode sulfate number concentration depending on the activation scheme used with MARC, but the fact that these differences do not show up in the zonal-average AOD suggests that MARC consistently predicts too-few coarse mode aerosol.

Present-day zonal average cloud macrophysical properties are summarized in Fig. 5b-d. MARC generally performs comparably with other models in re-producing zonal patterns in cloud fraction, cloud optical depth, and liquid water path. The *nenes*

and *PCM* schemes produce slightly higher cloud optical depth across all latitudes and particularly in the tropics relative to the other activation schemes. MARC tends to under-predict cloud optical depth and liquid water path in polar regions, although this error is common in nearly all AeroCom models as well. The preponderance of mixed-phase clouds greatly complicates assessing these regions, and MODIS retrievals can become unreliable due to changing surface conditions (in particular, reflective surfaces such as snow which increase in frequency towards the poles). MARC is well within the inter-model spread in

simulated cloud macrophysical properties across latitudes.

Following the discussion in Sect. 3.1.1 the largest activation-induced differences between simulations arises in cloud-top cloud droplet number (CDNC, Fig. 5e). Poleward of 60°, CDNC simulated by the *nenes* schemes is up to double that simulated by the *ARG* schemes. These differences are most pronounced in latitudes with significant anthropogenic aerosol emissions, particularly in the Northern Hemisphere. All the models shown here substantially under-predict CDNC in the extra-tropics,

but the MODIS-derived estimates are highly uncertain in this region (Bennartz, 2007). The combination of these differences in cloud microphysical properties yields small differences in the model-estimated shortwave cloud radiative effect (Fig. 5f).

### 3.2 PD-PI changes in clouds and radiation

Figure 7 illustrates that in absolute terms, difference in PD shortwave cloud radiative effect (CRE) simulated using each activation scheme is small. However, the change in cloud radiative effect between the PD and PI simulations (Fig. 7a) has

a spread of nearly $2\,\mathrm{Wm}^{-2}$ across all latitudes. Note that these differences can be much larger on local scales. Activation schemes which produce the smallest cloud-top droplet number concentration generally produce the largest differences in cloud radiative effect between the two emission scenarios. Relative to the *arg_comp* scheme, all other schemes produce a smaller cloud radiative effect in the PD emissions case (Fig. 7b); magnitude of these inter-scheme differences is comparable to the total





change between the PD and PI simulations for all schemes. The largest differences between schemes occur in the tropics and in the mid-latitudes of the Northern Hemisphere, both regions influenced by anthropogenic aerosol emissions and where the largest differences in CDNC also occur.

To better illustrate the sources of differences in simulated CRE, changes in aerosol and cloud microphysical properties between the PD and PI emission scenarios are shown in Fig. 6. Cloud condensation nuclei (CCN) robustly increase in Fig. 6a as aerosol emissions increase from the PI to the PD scenarios. Furthermore, CCN increases the most in the Northern Hemisphere, where anthropogenic aerosol emissions are the largest. The regions of largest increases in CCN also tend to feature the largest increases in CDNC (Fig. 6b), although there is a factor of 3-4 difference between CDNC simulated by the various activation schemes. At the same latitudes, droplet effective radius decreases (Fig. 6c), optical depth increases (Fig. 6d), and liquid water path increases (Fig. 6f). Changes in cloud fraction (Fig. 6e) are much noisier, but generally there is an increase in cloudiness between the two cases.

Regardless of which activation scheme is used, compared to Fig. 5c-d, liquid water path and cloud optical depth increase by up to 20% co-located where the largest increases in CDNC occur. The latitudes of the largest PD-PI differences in CCN, CDNC, droplet effective radius, and liquid water path are coincident with the largest changes in cloud optical depth and attendant shortwave CRE. Changes in cloud fraction do not necessarily coincide with these other changes in cloud properties, and instead maximize in the high-latitudes of the Northern Hemisphere. However, given that CCN (as a proxy for aerosol available to nucleate cloud droplets) differs little when using different activation schemes, the spatial pattern of these changes in cloud microstructure and CRE strongly suggests that the specifics of activation in the MARC simulations is driving the changes in CRE.

## 3.3 Influence on the aerosol indirect effect

Figure 8 shows differences in globally-averaged aerosol direct and indirect effects computed from the pairs of PD and PI runs for each activation scheme. The particulars of aerosol activation in these simulations produces differences of up to $0.8 \, \mathrm{W m^{-2}}$ between simulations, or a 100% increase by the strongest over the weakest estimate. Differences in the total aerosol influence on climate (, adopting the IPCC nomenclature "ERFaci+ari" to denote the separate contributions from the indirect (aci) and direct (ari) effects)(Boucher et al., 2013) here are strongly modulated by perturbations to the shortwave cloud radiative effect, ERFaci-shortwave, which is broadly consistent with the changes in cloud microphysical properties illustrated in Fig. 6.

Each perturbed component in the top-of-atmosphere (TOA) radiative budget are decomposed in Table 2. Consistent with expectations, there is relatively little variance in the direct radiative forcing diagnosed for each activation scheme, even accounting for the feedback of increased nucleation scavenging depressing aerosol number. Similarly, there is not much difference in the longwave CRE, which is dominated by ice-phase clouds and not directly influenced by adjusting the activation scheme. The longwave indirect effect in these models is net positive in all cases; dust and large sulfate particles in the model can nucleate ice crystals (Liu et al., 2007), and the increase in aerosol between the two emission scenarios yields ice clouds with modestly higher ice crystal number concentration, higher ice water path, and increased longwave influence relative to the shortwave. Us-





ing a similar modeling setup, Gettelman et al. (2012) previously showed a similar influence of aerosol on ice cloud-longwave radiation interactions.

Furthermore, we note that the indirect effect in the longwave is critically sensitive to the baseline ice crystal number burden simulated in the model. Additional tests using an alternative, aerosol-coupled ice nucleation scheme (DeMott et al., 2010)

decreased the longwave cloud radiative effect in MARC in a manner that scaled nearly linearly with ice crystal number. Using this alternative scheme produced much higher cloud-top ice crystal concentrations and ice water path, as well as a larger change between the two fields in the PI and PD simulations.

The majority of the difference in the indirect effect and net TOA radiative flux thus arises from changes in cloud interactions with shortwave radiation via cloud optical thickness. For the shortwave CRE alone, the spread between the different activation

schemes is larger than the net effect itself at $1.1\,\mathrm{Wm^{-2}}$. Both the longwave CRE and aerosol direct radiative forcing act to minimize the net radiative effect; in the simulations with the largest shortwave contribution, both the direct and cloud longwave effects are smaller and larger, respectively. The small spread in direct effect in these simulations correlates very strongly with the change in global-average aerosol optical depth, but changes in that field are only loosely related to changes in the available CCN

## 3.4   Indirect effect sensitivity to aerosol-cloud perturbations

We highlight in Fig. 9 the relationship between the change in shortwave CRE to model-simulated aerosol burdens over maritime and continental regions. The increase in direct aerosol and precursor gas emissions in the PD emission case leads to an increase in both aerosol optical depth (AOD) and the availability of cloud condensation nuclei (CCN). However, the exact magnitude of this increase is dependent on the formulation of the aerosol module in each model, especially their simulated size distributions.

Inter-model spread in the PD case for AOD and its sensitivity to perturbation from the PI climate has been associated with up to a $0.5\,\mathrm{Wm^{-2}}$ spread in estimates of the direct effect (Shindell et al., 2013), but it also has implications for the indirect effect following Fig. 9a.

The largest AOD increases occur over land, and these are also associated with larger perturbations to shortwave CRE. The MARC simulations and most of the AeroCom models do not simulate major increases in AOD over the ocean, even while there

is considerable spread in the magnitude of ΔCRE in that regime. One potential explanation for the small response in AOD over these ocean regimes could be that AOD is dominated by large natural aerosol such as sea salt in these regions, which would not directly increase in response to anthropogenic emissions.

CCN also directly increases with anthropogenic emissions. However, in contrast with AOD, small PD-PI changes in CCN are associated with a larger (more negative) indirect effect (Fig. 9b). For the entire set of AeroCom models and the MARC

simulations performed here, as the model-simulated ΔAOD increases, the indirect effect becomes weaker. The slope of the ΔCCN-ΔCRE relationship is much steeper for the MARC simulations than the AeroCom ones (due to an outlier model with relatively insensitive CCN fields. Different aerosol metrics have previously been shown to have different relationships with model- and satellite-derived estimates of the indirect effect, but they usually have the same sign (Penner et al., 2011). This suggests that each metric is capturing a different facet of the aerosol size distribution which may or may not be relevant to





changes in the indirect effect, depending on how they influence CDNC, which would potentially be conditioned on the initial climatology of CDNC simulated under a PI emissions case.

To assess this influence, we plot similar relationships between CDNC, liquid water path (LWP) and liquid cloud fraction in Fig. 10. The spread change in CDNC between different models very weakly correlates with the strength of the indirect effect
(Fig. 10a). Instead, a much better predictor of the indirect effect is the pre-industrial CDNC (Fig. 10b), which itself strongly positively correlates with the change in CDNC between PI and PD. This is evident in Fig. 10-b, which shows a negative relationship between the pre-industrial CDNC and the indirect effect (negative in the sense that the magnitude of $\Delta$ CRE is decreasing as PI CDNC is increasing) The models which produce higher CDNC for the same background or natural aerosol tend to have weaker indirect effects.

Liquid water path and cloud fraction exhibit a different relationship with the indirect effect (Fig. 10c-f). Using different activation schemes in MARC directly influences the sensitivity of LWP to aerosol perturbation through enhancement of CDNC, which strongly modulates the indirect effect. Although MARC generally simulates much larger LWP in the PI case, the range of indirect effects it simulates spans the spread of those obtained from the AeroCom models. The same relationships hold for liquid cloud fraction, which is correlated with LWP in both MARC and the ensemble of AeroCom models, particularly for
oceanic regimes. The AeroCom models simulate far more diversity in LWP and cloud fraction in the PI case, but tend to agree on the magnitude of change PI and PD, as does MARC.

Using different activation cases produces larger differences in simulated PI CDNC versus either LWP or cloud fraction. This suggests that the large-scale cloud properties in MARC are insensitive to the background aerosol level. Instead, changes in the simulated indirect effect arising from the different activation schemes is dominated by the first indirect effect and a change in
the ambient CDNC burden, which is driving microphysical changes leading to the observed perturbations in both cloud optical properties and their spatio-temporal distribution.

## 3.5  Summarizing the influence of aerosol activation

To contextualize the influence of aerosol activation on the indirect effect in the simulations presented here, we plot estimates of the indirect effect (ERFari+aci) reported by Boucher et al. (2013) in Fig. 11. These include a highlighted subset of models
and results combining satellite observations with model analysis (Fig. 11, AR5 Table 7.4), results from a previous model inter-comparison using CMIP-class models (Fig. 11, AR5 Table 7.5 Shindell et al., 2013), and new estimates derived from the AeroCom models considered here and the various configurations of MARC with different activation schemes. The estimates presented here span a wide variety of potential model physics and aerosol couplings, and therefore different aerosol indirect effects.

In our simulations with MARC, differences in aerosol activation produce a spread in estimates of the indirect effect comparable in magnitude to the total inter-model diversity. Furthermore, our estimates–especially for the configurations with lower CDNC–tend to cluster in the higher-end of estimates compared to previous inter-comparisons. The same is true for the AeroCom models considered here, although we note that four of the AeroCom models are closely-related variants of the same parent model as MARC (the NCAR CAM5.3) and therefore the estimates are not totally independent of one another.





Our range of indirect effects induced by different activation treatments is much larger than the few others reported in the literature. By re-ordering the droplet activation calculation in each model timestep, Gettelman (2015) induced a $0.43\,\mathrm{Wm}^{-2}$ decrease in the magnitude of the indirect effect; Ghan et al. (2011) reduced it by just $0.16\,\mathrm{Wm}^{-2}$ when switching between two different activation schemes. This range is much smaller than the sensitivity of $0.86\,\mathrm{Wm}^{-2}$ we report for the experiments

conducted here. We note that both of those previous estimates of the sensitivity of the indirect effect to activation used nearly identical global models (early versions of the NCAR CAM5 with the same aerosol module). Our use of a unique aerosol model could contribute to some of the difference in the range of estimates of the indirect effect. This possibility can indirectly be tested using the suite of model results presented in this work, since one included AeroCom model is the NCAR CAM5.3 in its default configuration, which should be nearly identical to the *arg_comp* MARC configuration here save for the different

aerosol module. The difference in ERFari+aci between these two simulations is $0.45\,\mathrm{Wm}^{-2}$, which is half of the total range reported here for MARC with different activation schemes.

## 4   Discussion and Conclusions

In this study, we have quantified the influence of the representation of droplet activation in global models on the sensitivity of the aerosol indirect effect. Using a suite of state-of-the-science activation parameterizations incorporated into our global aerosol-

15 climate model, MARC, we performed simulations under both pre-industrial and present-day aerosol emissions scenarios to estimate the magnitude of the indirect effect and its relationship to changes in both cloud and aerosol fields. Previously, few studies exploring the indirect effect focused explicitly on the role of droplet activation, instead concentrating on either the processes that produce ambient aerosol itself (emissions and atmospheric chemistry) or the results of changes occurring purely in cloud droplet number concentration (such as imposed minimum values for cloud droplet number or in microphysical

processes which modify it). Beyond assessing three unique activation schemes, we supplement our analysis by considering three additional, idealized droplet activation schemes which use a heuristic to simplify accounting for competition between different aerosol modes for moisture during the nucleation process.

The relationship between cloud droplet number concentration and aerosol in MARC is critically influenced by the representation of droplet activation. Estimates of CDNC in the present-day climate are up to 40% higher in polluted regimes when

using the most-sensitive activation scheme, and the increase from pre-industrial to present day is up to twice as large. CDNC in regimes dominated by natural aerosol, especially remote marine regions with prevalent sea salt, is also impacted by the activation scheme. Using the advanced droplet activation schemes included here, which explicitly account for biases due to giant CCN particles, helps reduce the under-prediction in maritime regimes compared to satellite observations and in polluted regimes compared to in situ observations. However, MARC systematically produces too few CDNC in most parts of the globe.

While this could be due to misrepresentation of aerosol-cloud processes, we emphasize that it could also be fundamentally related to the simulated aerosol size distribution within MARC and how it apportions aerosol number and mass in the size ranges where likely-CCN reside. However, evaluations of previous versions of the model (Wang, 2004; Kim et al., 2008; Ekman et al.,




2012) and those presented in the Supplement to this work suggest that MARC captures the bulk aerosol climatology rather well across the globe; thus, these systematic biases in CDNC and CCN could apply to other aerosol-climate models as well.

Compared to available satellite measurements and the models participating in the AeroCom inter-comparison, though, MARC does well at capturing the present-day climatology of cloud and radiation fields, likely because its parent model,
the NCAR CESM, itself is well-tuned towards this end. However, the details of activation and how it influences cloud microphysics plays a major role in setting the shortwave cloud radiative effect. Under present-day emissions, the differences between that effect for each of the different activation schemes is as large as the change from the pre-industrial case for each scheme. This leads to large differences in the modeled indirect effect in each model, almost entirely occurring due to the shortwave cloud radiative effect. The resulting spread in indirect effect estimates is twice as large as that previously reported by studies
considering activation, and about as large as the inter-model spread from both historical and recent model inter-comparisons, which consider models including a variety of different aerosol effects.

We note that the pre-industrial CDNC burden is a very strong predictor of the strength of the indirect effect, but not necessarily the change between pre-industrial and present day; this hints at the previously-hypothesized buffering effect of clouds on aerosol perturbations (Stevens and Feingold, 2009). Our analysis strongly supports this notion; our case which produces–by
far–the largest change in CDNC yields nearly the smallest indirect effect, but simultaneously produces the highest values for pre-industrial CDNC. While consistent with the results of Hoose et al. (2009), this is opposite of the results reported by Storelvmo et al. (2009), although their model uses prescribed aerosol fields so there is no interaction between cloud and aerosol processes. Changes in liquid water path and cloud fraction correlate strongly with modeled total changes in cloud radiative effect, but poorly with changes in CDNC in our simulations. This suggests that the buffering effect must be dominated by changes
in the second indirect effect, rather than the first. Both of these relationships hold for the broader sample of models provided by the AeroCom inter-comparison.

The weight of these results suggests an important role for activation in setting the sensitivity of the indirect effect. However, we caution that our approach is not able to disentangle the influence of activation from that due to the underlying aerosol model and its implicit aerosol size and CCN distributions. This is not meant to diminish the influence of cloud microphysical
treatments on the indirect effect; Gettelman (2015) illustrates the importance of the implicit cloud lifetime effects arising from liquid water path changes associated with aerosols in contributing to the indirect effect. But since these relationships are themselves highly sensitive to simulated CDNC, the influence of the aerosol size distribution and activation is somewhat more fundamental and just as poorly constrained by available observations. Furthermore, because of the chain of sensitivities initiating with the aerosol size distribution and activation, estimates of the indirect effect produced from models with highly-
simplified aerosol-CDNC relationships (such as explicitly prescribed CDNC or empirical fits to aerosol mass or volume) are likely significantly biased.

To test this idea, additional work following this and Kodros and Pierce (2017) where the embedded aerosol model in a given global model is substituted while all other physics remain the same could prove useful. We also suggest that future sensitivity analyses in the vein of Carslaw et al. (2013) include perturbations to the fundamental activation or CCN-CDNC relationship to
account for this source of uncertainty. Constraining this uncertainty is a different matter altogether. Current observations can





not constrain the spatio-temporal variance in the ambient aerosol size distribution, which is critical in setting the sensitivity of CDNC and cloud optical properties to aerosol perturbations. New data from aircraft sampling clouds in regimes with the greatest aerosol-cloud sensitivities across the globe could play a key role in addressing this limitation.

In order to better understand contemporary and account for future climate change, the aerosol community must continue to seek constraints on the aerosol indirect effect. Although epistemic uncertainty due to unknown pre-industrial emissions complicates this task, the role of droplet activation illustrated in this work highlights an additional path that the community may explore to provide indirect or emergent constraints on AIE via the basic aerosol-CDNC relationship.

*Code and data availability.* A Git repository archiving the scripts and build files used to process the MARC and AeroCom output and perform the analyses presented in this work can be found at https://github.mit.edu/darothen/aerocom_activation; documentation on which
scripts and notebooks perform which analyses can be found in the README.md file therein. The source code for MARC can be found in a Git repository at https://github.mit.edu/marc/marc_cesm, as well as instructions for setting up the model from a standard CESM installation. MARC v1.0.2, used in this work, is archived with DOI 10.5281/zenodo.168192. The emissions datasets and scripts used to generate them for this work are archived at https://github.mit.edu/marc/marc_input.

Output from the simulations used in this analysis are available upon request.

**Appendix A: Droplet Nucleation and Activation Schemes**

Droplet nucleation, or aerosol activation, refers to the process through which aerosol, which are entrained through the base or sides of a cloud, grow into a nascent cloud droplet population. Assessing this process is complicated by the fact that latent heat release from condensation on the surface of aerosol within adiabatically ascending (and therefore cooling) parcel provide a strong feedback, limiting the development of supersaturation (relative humidity over 100%) and thus the potential for some
particles (usually referred to as cloud condensation nuclei, or CCN) to grow into droplets. Contrary to its common usage in the field, CCN is not necessarily a stand-alone, diagnostic measure of a given aerosol population; instead, *all* aerosol are potentially CCN, given an updraft sufficient enough in strength to drive a high-enough supersaturation such that they grow large enough to activate. In the ensuing discussion, we eschew the term CCN and instead focus explicitly on total aerosol number ($N_a$) and cloud droplet number ($N_d$), emphasizing the importance of the activation process in determining how many particles will
nucleate a droplet.

The aerosol size distributions predicted by MARC are explicitly used to constrain droplet activation in the stratiform cloud microphysics scheme; the shallow and deep convection schemes do not include prognostic droplet number. With respect to stratiform clouds, activation is driven by a characteristic sub-grid scale vertical velocity derived from the turbulent kinetic energy





predicted by the University of Washington shallow convection and moist turbulence parameterization (Park and Bretherton, 2009):

$$\hat{w} = \max\left(\sqrt{\frac{2}{3} \times \mathrm{TKE}}, 0.2\right) \mathrm{ms}^{-2} \tag{A1}$$

All of the aerosol species described in Table 1 — except for the pure BC and OC modes — are included in droplet activation calculations. Each mode is assigned a fixed hygroscopicity (Petters and Kreidenweis, 2007), except for the mixed sulfate-organic carbon (MOS) mode, for which the hygroscopicity is computed as a volume-weighted mean based on the amount of each species present. Dust is assumed to be comprised of weakly-hygroscopic minerals following Scanza et al. (2014); the mixed sulfate-black carbon mode (MBS) particles are assumed to have a surface area purely comprised of sulfate, which dictates their hygroscopicity.

In this work, we have implemented several additional activation schemes and associated variants. Fundamentally, each activation scheme attempts to simplify the calculation of the maximum supersaturation achieved in a parcel under the simultaneous influence of both cooling form adiabatic ascent and warming from latent heat release as water condenses on particles contained within the parcel. The total of this physical process can be summarized in a single, integro-differential equation:

$$\alpha V = \gamma G S_{\mathrm{max}} \int_0^{S_{\mathrm{max}}} \left(r^2(t_{\mathrm{act}}) + 2G \int_{t_{\mathrm{act}}}^{t_{\mathrm{max}}} S dt\right)^{1/2} \frac{dN}{dS_c} dS \tag{A2}$$

Here, $\alpha$ and $\gamma$ are functions weakly dependent on the parcel's temperature and pressure, $V$ is the velocity of the parcel's adiabatic ascent, $r^2(t_{\mathrm{act}})$ indicates the size of a given parcel at the time it activates (grows large enough that, following Köhler theory, further condensational growth is thermodynamically favorable even if the relative humidity drops, Seinfeld and Pandis (2006)), $G$ is a particle-dependent condensational growth coefficient, $t_{\mathrm{max}}$ indicates the time at which the maximum supersaturation achieved in the parcel, $S_{\mathrm{max}}$, occurs, and $\frac{dN}{dS_c}$ represents the aerosol size-distribution re-written in terms of Köhler theory and expressed as a function of a particle's "critical" size. For a rigorous derivation of Eq. A2 and discussion on the assumptions necessary to simplify it, we refer the reader to Ghan et al. (2011).

It is immediately useful to simplify Eq. A2 by partitioning the integral over $dS$ into two ranges, $[0, S_{\mathrm{part}}]$ and $[S_{\mathrm{part}}, S_{\mathrm{max}}]$. These ranges effectively split the aerosol population into a subset which do not grow very much between $t_{\mathrm{act}}$ and $t_{\mathrm{max}}$ and one which does. Partitioning the integral in this produces the alternative formulation:

$$\alpha V = \gamma G S_{\mathrm{max}} \left[\int_0^{S_{\mathrm{part}}} \left(2G \int_{t_{\mathrm{act}}}^{t_{\mathrm{max}}} S dt\right)^{1/2} \frac{dN}{dS_c} dS + \int_{S_{\mathrm{part}}}^{S_{\mathrm{max}}} r(t_{\mathrm{act}}) \frac{dN}{dS_c} dS\right] \tag{A3}$$

The first two activation schemes employed in this study utilize this formulation of the activation equation; the third implicitly uses the equation, albeit in an alternative form as a system of coupled ordinary differential equations.





## A1  Abdul-Razzak and Ghan (2000) (*ARG*)

Abdul-Razzak et al. (1998) assume the aerosol size distribution can be described by a single lognormal mode, and parameterized the dependence between $S_{max}$ and the critical supersaturation of the geometric mean radius of that mode, $S_m$. This parameterization yields two non-dimensional terms, $\eta_m$ and $\zeta$. The mode geometric mean standard deviation, $\sigma_g$, does not appear in either term, allowing Abdul-Razzak and Ghan (the version used here; *ARG*) to use a parcel model to relate $S_{max}$ to $S_m, \eta_m, \zeta$, and $\sigma_g$. Although originally tuned assuming a condensation coefficient ($\alpha_c$) of unity, Ghan et al. (2011) proposed scaling $G$ in the expressions for $\eta_m$ and $\zeta$ to account for smaller values. However, within the scheme, $G$ is determined in such a way that the *ARG* scheme neglects gas kinetic effects completely and cannot be easily extended to account for organic films or surfactants, or other effects which may module surface tension or $\alpha_c$ on a particle-by-particle basis.

## A2  Morales Betancourt and Nenes (2014b) (*nenes*)

Nenes and Seinfeld (2003) developed an iterative scheme to solve Eq. A2 based on a sectional aerosol distribution, which Fountoukis and Nenes (2005) extended to accommodate lognormal modes. The iterative nature of its computation allows the scheme to branch under two conditions. In the first, when kinetic limitations on droplet growth are expected, $S_{math}$ and $S_{part}$ are parameterized empirically from a suite of pre-computed numerical simulations. In the alternative branch, the integrals in Eq. A2 are analytically computed with the assumption that $r^2(t_{act}) \gg 2G \int_{t_{act}}^{t_{max}} S dt$; in other terms, that particles which activate do not grow much beyond their critical size. The resulting set of equations is similar to the one derived by Abdul-Razzak and Ghan (2000), owing to their common starting point and lognormal mode assumptions.

Owing to its flexibility, the iterative scheme has been successively modified over the course of several follow-on papers. Barahona and Nenes (2007) incorporated the effect of entrainment on a parcel in which activation occurs; Kumar et al. (2009) added the ability to include insoluble CCN using an adsorption framework; Barahona et al. (2010) modified the original equations to comprehensively account for the impact of giant CCN; and Morales Betancourt and Nenes (2014a) (the version used here; *nenes*) revised the population-splitting approach underlying the iterative scheme to better handle giant CCN and avoid an unphysical discontinuity in their branching conditions. Critically, these schemes improve on the *ARG* scheme by accounting for the interplay between gas kinetics and the diffusivity of water vapor. To accomplish this, the schemes compute $G$ using an average value of the diffusivity over a particle size range corresponding to the mode undergoing activation. This obviates the need for scaling $G$ based on the chosen value of $\alpha_c$.

However, there are two drawbacks to this method. First, the need for a reference parcel model for tuning is not completely eliminated; the branch of of the iterative scheme accounting for kinetic limitations still requires an empirical relationship. Second, the iterative scheme can be much more computationally expensive than the *ARG* scheme, since the calculations–including costly error function evaluations–must be performed multiple times.





## A3 Rothenberg and Wang (2016) (*PCM*)

An alternative approach to parameterizing droplet activation calculations involves building look-up tables for inclusion in global models. However, this approach is not widely used; in a modern model, the parameter-space influencing activation is very large and covering such a space in a look-up table is intractable. Fundamentally, a look-up table is a cache of results

from a higher-complexity model (in this case, a detailed parcel model), which are used to generate a piecewise-planar response surface on the fly during model run-time. With respect to activation, look-up table emulation has been successfully employed with cloud-resolving models featuring simple aerosol/CCN distributions (Ward et al., 2010; Ward and Cotton, 2011) and to improve the performance of activation schemes in global models (Ming et al., 2006).

Rothenberg and Wang (2016) applied polynomial chaos expansion to emulate a detailed parcel model simulating activation

of a single lognormal aerosol mode. The technique was later extended to analyze and assess activation of a full-complexity aerosol scheme (Rothenberg and Wang, 2017). These schemes provide a means to efficiently embed the physics of a full-complexity parcel model within the context of a global model. The *PCM* scheme provides a polynomial function mapping a set of pre-determined input parameters (such as the parcel updraft speed, temperature/pressure, and the moments closing the aerosol size distribution) to a value of $S_{\mathrm{max}}$, which can then be used to diagnose equilibrium droplet activation. Like a look-up

table, a cache of parameters (the coefficients of the mapping chaos expansion basis polynomials) must be pre-computed to use the *PCM*. However, the highest-order expansion of the 8 terms reported in Rothenberg and Wang (2016) required just 495 coefficients to be saved to disk, compared to an isotropic or full-factorial design look-up table which would require $10^8$ values for the same setup.

Unlike the *ARG* and *nenes* schemes, feedbacks on the development of $S$ in a cloudy parcel due to kinetic limitations on

droplet growth are explicitly treated in the *PCM* scheme, as are the dependence of condensation ($G$) on particles' composition and size. However, inertial limitations on activation (where a particle may experience its critical supersaturation, but has not yet grown large enough to explicitly activate) remain a source of bias in the formulation applied here, because of the equilibrium assumption used to diagnose $N_d$ from $S_{\mathrm{max}}$.

## A4 Minimum-$S_{\mathrm{max}}$ Heuristic

Additionally, we supplement the *PCM*, *nenes*, and *ARG* schemes with an alternative formulation for expressing the competition between multiple aerosol modes to nucleate droplets. Rothenberg and Wang (2017) showed that in the majority of cases, a single mode "dominated" activation calculations; that is, there was one aerosol mode which, when activated alone, in isolation of its competitor modes, provided a strong constraint activation dynamics and accurately predicted $S_{\mathrm{max}}$ within a few percent. This "dominant" mode is defined as the mode which, in isolation of the other modes, produces the smallest $S_{\mathrm{max}}$ when present

in an adiabatically cooling air parcel. Using just the "dominant" mode as a surrogate for the complete aerosol population will always lead to an over-prediction of $N_d$ because neglecting additional modes effectively limits the surface area of the aerosol available for condensation, and thus the potential for a large source of latent heat release to overcome adiabatic cooling in the parcel and limit the development of $S_{\mathrm{max}}$; this effect tends to dominate the reduction in the sink of ambient water vapor which



typically constraints $S_{\max}$. However, a large ensemble of calculations showed this over-prediction of $N_d$ to be small in most situations.

   While no substitute for detailed calculations of droplet activation which take into account all potential factors, using just the "dominant" mode for assessing activation provides a simple heuristic for widening the pool of potential activation schemes to

couple in our model. For the *ARG* and *nenes* schemes, the heuristic is applied by looping over each available aerosol mode and computing activation independently. For the *PCM* schemes, the same technique applies, but we use a chaos expansion explicitly built for single mode calculations. Including the minimum-$S_{\max}$ scheme variants in our global modeling calculations allows us to explore the dependence of the indirect effect on the nuances of droplet activation without resorting to imposing unphysical restrictions on either cloud droplet number or aerosol activation dynamics.

*Competing interests.* The authors declare that they have no conflict of interest.

*Acknowledgements.* The work in this study was supported by the National Science Foundation Graduate Research Fellowship Program under both NSF grant 1122374 and NSF grant AGS- 1339264, the National Research Foundation of Singapore through the Singapore–MIT Alliance for Research and Technology and the interdisciplinary research group of the Center for Environmental Sensing and Modeling, and the U.S. Department of Energy, Office of Science (DE-FG02-94ER61937). We thank Steve Ghan (PNNL) and Athanasios Nenes (Georgia

Tech) for reference implementations of their activation parameterizations, and Natalie Mahowald (Cornell) for assistance in tuning the natural dust simulations in MARC.



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





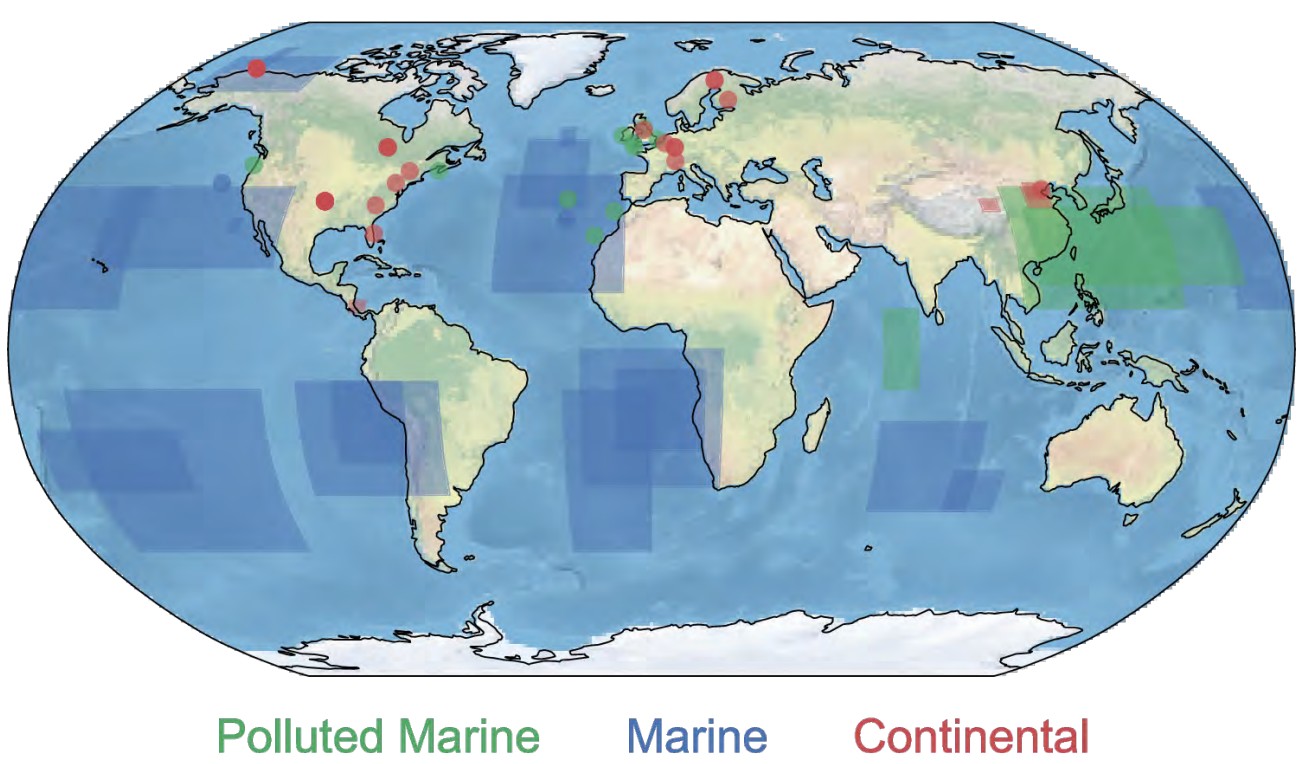

**Figure 1.** Locations of *in situ* observational data reported by Karydis et al. (2011). Different colors correspond to classifications of different aerosol regimes.

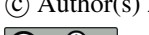


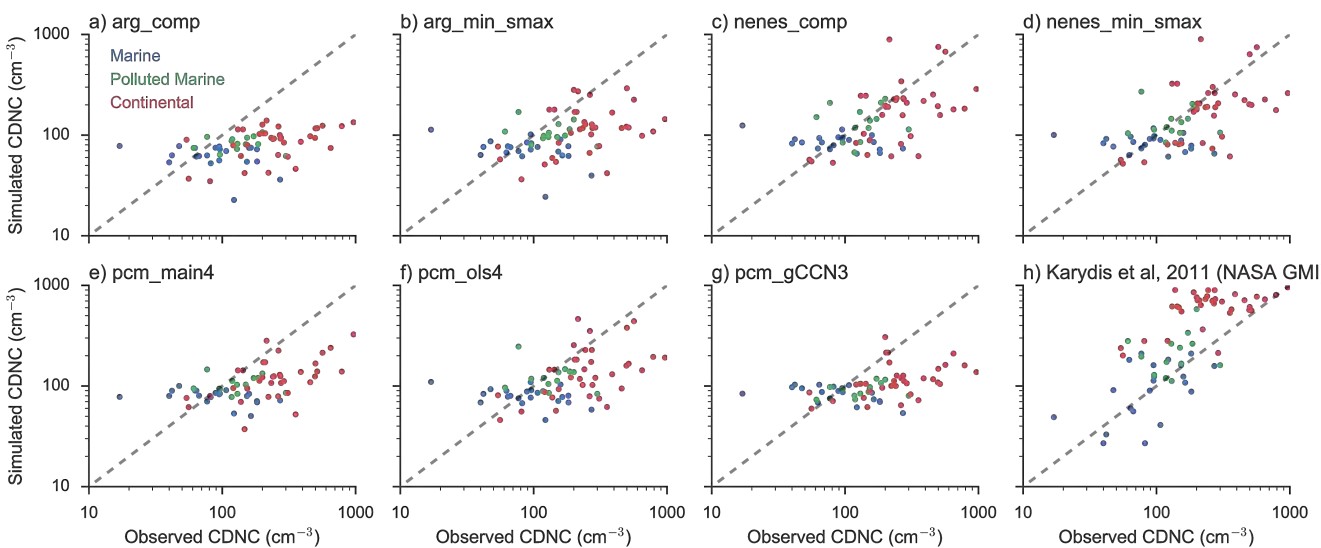

**Figure 2.** One-to-one comparisons between observed and simulated cloud droplet number concentrations from regions across the globe. Panels (a)-(g) show results from the MARC simulations using the indicated droplet activation schemes; panel (h) shows results from CDNC modeled by a chemical transport model with detailed aerosol and activation treatments.



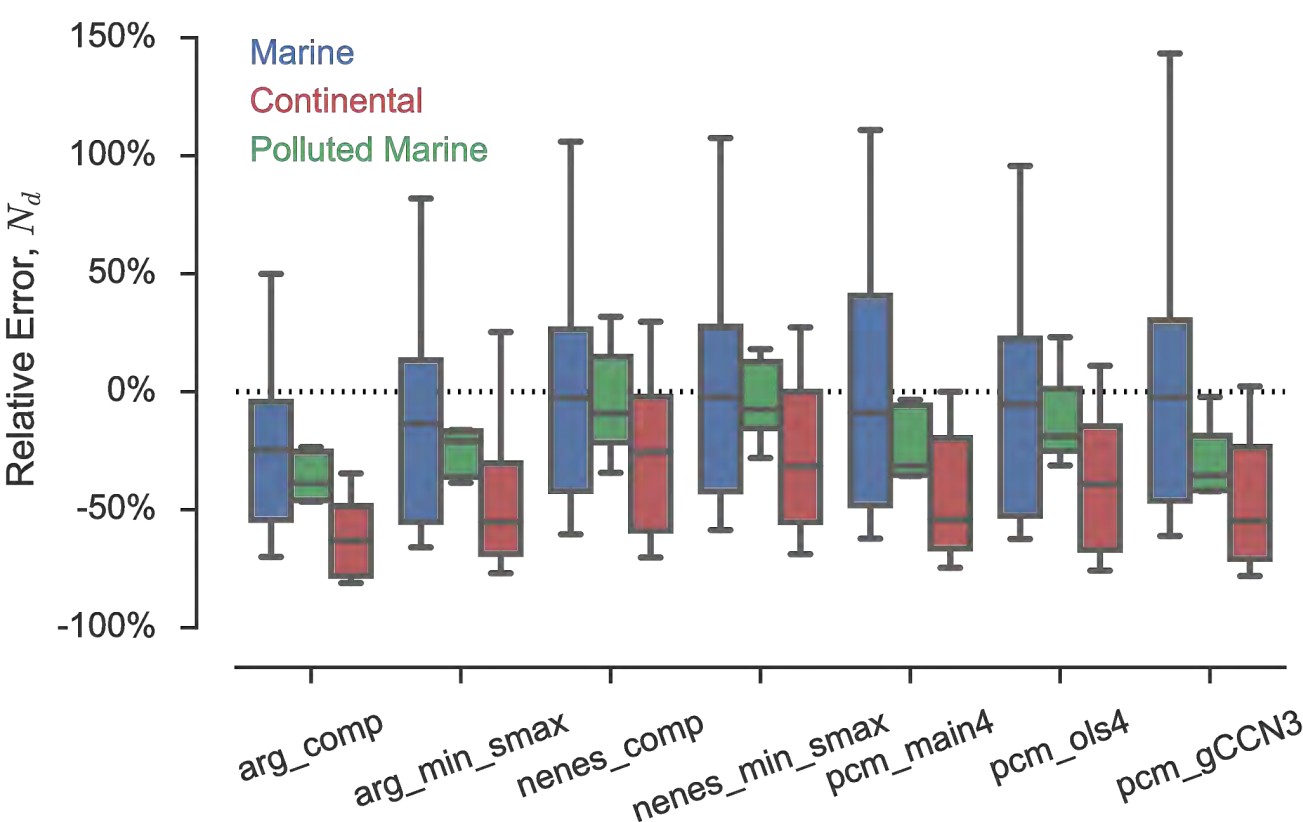

**Figure 3.** Distributions of relative error between observed and simulated CDNC ($N_d$) for each configuration of MARC, aggregated by region.





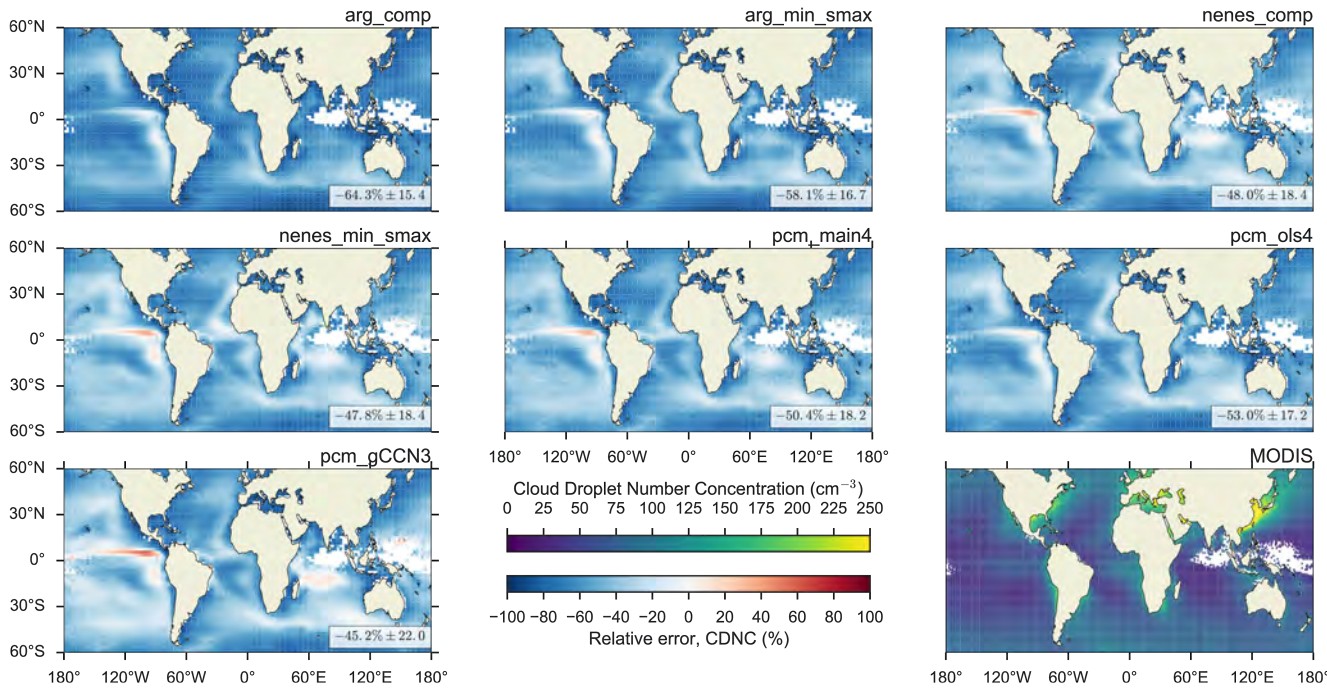

**Figure 4.** Global distribution in relative error of MARC-simulated CDNC versus MODIS-derived satellite observations (bottom-right)





**Figure 5.** Zonal average aerosol, cloud, and radiation fields under present-day emissions scenario. Colored lines correspond to configurations of MARC using different activation schemes; black lines are derived from CERES-EBAF (SW Cloud Radiative Effect) and MODIS (all other panels) observations. The shaded gray area corresponds to the inter-model spread for all available models participating in the AeroCom Indirect Effects Experiment; dashed white lines are the zonal averages for each participating model. Cloud droplet number is computed at cloud-top, using only grid cells over the ocean. Here, SW Cloud Radiative Effect is computed using the difference between clear-sky and all-sky fluxes.





**Figure 6.** Same as Fig. 5, except illustrating differences in indicated fields between pre-industrial (PI) and present-day (PD) emissions scenarios. CCN here is computed at an altitude of 1 kilometer in the model.





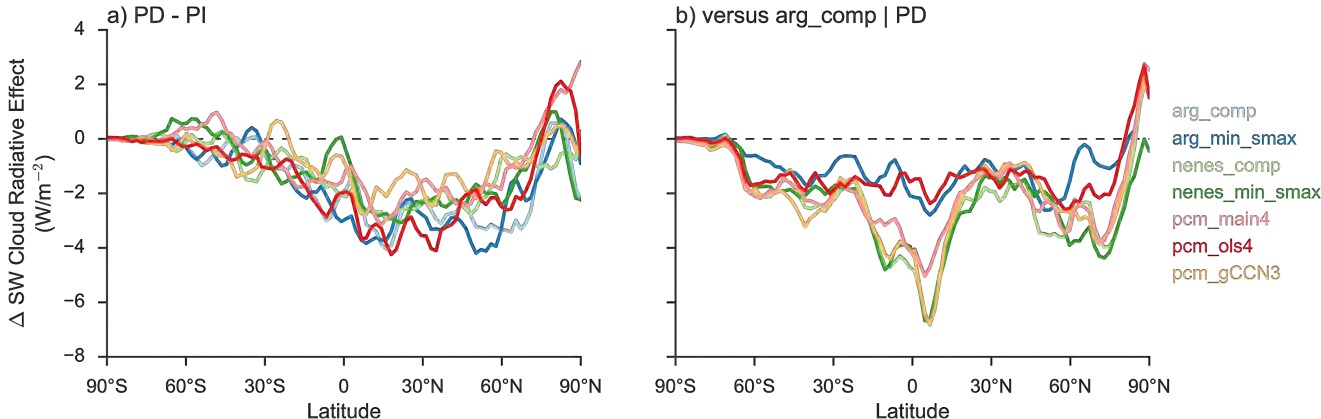

**Figure 7.** Difference in shortwave cloud radiative effect between pre-industrial and present-day emissions scenarios (a) and relative to the arg_comp simulation for present-day emissions (b).

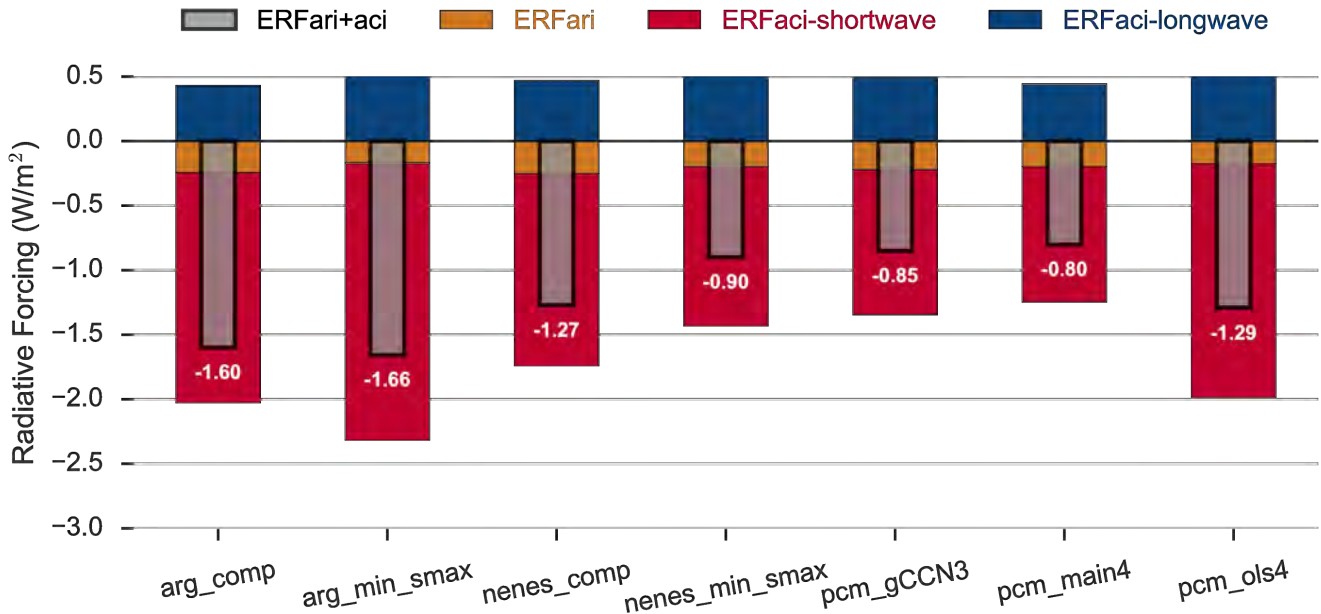

**Figure 8.** Global-average effective radiative forcing for aerosol direct radiative effects (ERFari) and indirect effects (ERFaci) in both the shortwave and longwave. The total effect is computed as the sum of the direct and both indirect components.





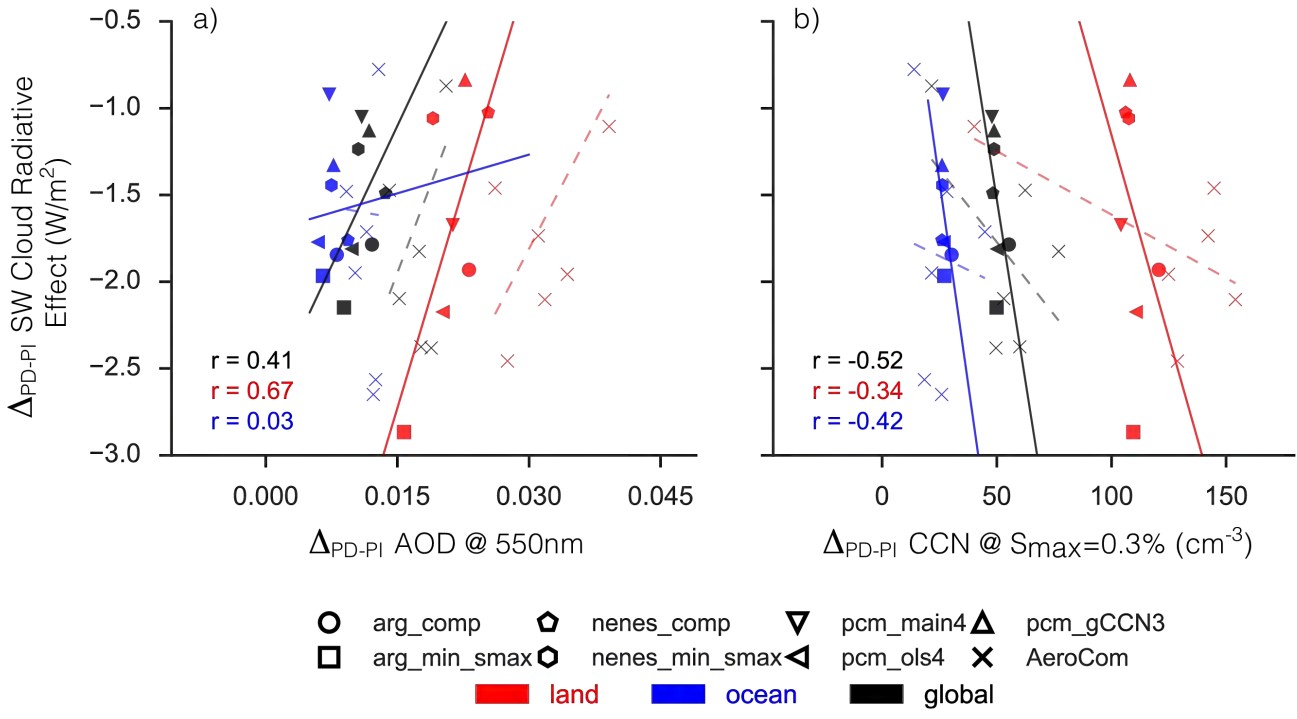

**Figure 9.** Relationship between regionally-averaged PD-PI change in shortwave cloud radiative effect and aerosol optical depth (a) or CCN concentration (b). Colors denote averages over all land area (red), ocean (blue), or whole globe (black). Glyphs denote MARC simulations with different activation schemes; X denotes AeroCom model. Linear regressions for MARC simulations are represented by solid lines, with corresponding correlation coefficient indicated on plot; linear regressions for AeroCom models are given by the dashed lines.



**Figure 10.** Similar to Fig. 9; top row (a-c) denotes relationship between PD-PI changes in shortwave cloud radiative effect and cloud properties, bottom row (d-f) shows regression versus averages from PI emissions scenario.





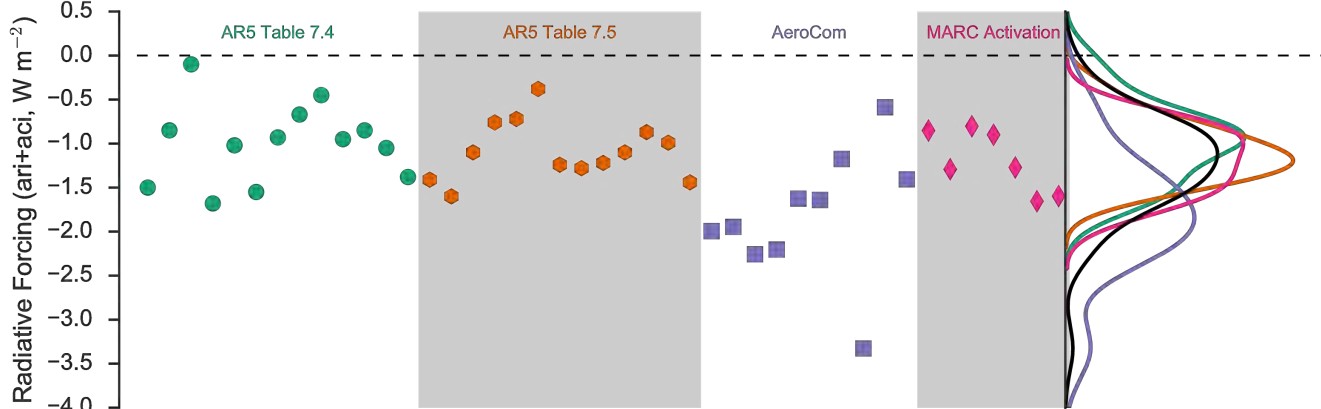

**Figure 11.** Comparison between estimates of ERFari+aci derived here and a subset of those previously reported by the IPCC AR5. Corresponding kernel density estimates of the distribution of ERFari+aci are given on the right-hand panel; the solid black curve shows the distribution accounting for all the estimates on the plot.

| Aerosol Mode | Geometric Mean Particle Diameter (μm) | Geometric Std Deviation ($\sigma_g$) | Density (g cm$^{-3}$) | Hygroscopicity ($\kappa$) |
|---|---|---|---|---|
| NUC | 0-0.00584 | 1.59 | 1.8 | 0.507 |
| AIT | 0.00584-0.031 | 1.59 | 1.8 | 0.507 |
| ACC | >0.031 | 1.59 | 1.8 | 0.507 |
| OC | - | 2.0 | 2.0 | $10^{-10}$ |
| MOS | - | 2.0 | † | † |
| BC | - | 2.0 | 2.0 | $10^{-10}$ |
| MBS | - | 2.0 | 2.0 | 0.507 |
| DST01 | 0.16 | 1.4 | - | 0.14 |
| DST02 | 0.406 | 1.4 | - | 0.14 |
| DST03 | 0.867 | 1.4 | - | 0.14 |
| DST04 | 1.656 | 1.4 | - | 0.14 |
| SSLT01 | 0.5 | 1.59 | - | 1.16 |
| SSLT02 | 2.0 | 1.37 | - | 1.16 |
| SSLT03 | 5.0 | 1.41 | - | 1.16 |
| SSLT04 | 15.0 | 1.22 | - | 1.16 |

**Table 1.** MARC aerosol mode size distribution and chemistry parameters. The MOS mode (†) has a composition-dependent density and hygroscopicity which is computed using the internal mixing state of organic carbon and sulfate present at a given grid-cell and timestep.



| Activation Scheme | $\Delta$ R | Total CRE | $\Delta$ SW CRE | $\Delta$ LW CRE | DRF | $\Delta$ RF | Res |
|---|---|---|---|---|---|---|---|
| arg_comp | -1.66 | -1.35 | -1.79 | 0.43 | -0.24 | -1.6 | 0.07 |
| arg_min_smax | -1.62 | -1.49 | -2.15 | 0.66 | -0.17 | -1.66 | -0.04 |
| nenes_comp | -1.35 | -1.02 | -1.49 | 0.47 | -0.25 | -1.27 | 0.07 |
| nenes_min_smax | -0.9 | -0.70 | -1.24 | 0.53 | -0.20 | -0.9 | -0.001 |
| pcm_gCCN3 | -0.72 | -0.63 | -1.13 | 0.50 | -0.22 | -0.85 | -0.13 |
| pcm_main4 | -0.85 | -0.61 | -1.05 | 0.45 | -0.20 | -0.8 | 0.05 |
| pcm_ols4 | -1.32 | -1.12 | -1.81 | 0.70 | -0.18 | -1.29 | 0.03 |

**Table 2.** Aerosol direct and indirect effects (in $Wm^2$) for the different activation schemes considered in this study. In all cases, we consider the change in the top-of-atmosphere radiative flux to compute these metrics (the net balance of which is given by $\Delta$ R). Our decomposition of the shortwave indirect effect (SW CRE) follows Ghan (2013) to account for above-cloud scattering and absorbing aerosol; similarly, the direct effect is computed diagnostically within each simulation. Following Gettelman (2015) we compute a residual between the top-of-atmosphere radiative imbalance and the direct and indirect effects such that Res = Total CRE + DRF - $\Delta$ R