# Peer review of "On the representation of aerosol activation and its influence on model-derived estimates of the aerosol indirect effect"

_Atmospheric Chemistry and Physics, 2017_

## Referee Comment (RC1) · Anonymous Referee #1 · 8 Oct 2017

**Review of Rothenberg et al. for Atmospheric Chemistry and Physics**

*General Comments*

In "On the representation of aerosol activation and its influence on model-derived estimates of the aerosol indirect effect", Rothenberg et al. integrated multiple droplet activation schemes into the Community Earth System Model (CESM) with a well-established, more-detailed aerosol module to quantify the influence of these parameterizations on the calculated aerosol indirect effect (AIE). The aerosol module tracks the evolution of number and mass for lognormally-distributed sulfate (three modes), black carbon, organic carbon, mixed modes, dust, and sea salt. The aerosol and droplets are coupled with radiative transfer and cloud microphysics, which are representative of stratiform clouds. The authors integrate three activation schemes as well as derivations of each scheme that employ a heuristic for more quickly calculating activation based on the dominant mode. They evaluate the predicted cloud droplet number concentration (CDNC) against in situ and satellite-based observations and demonstrate that these observations are insufficient for identifying a singular scheme as successful. Then, they comprehensively and clearly assess the AIE, in which they find a doubling of the strongest compared with the weakest result, and the parameters influencing AIE. Comparing the dependence of the change in cloud radiative effect (CRE) on other parameters and the degree of spread in the AIE in these simulations to the range of AeroCom and IPCC, the choice of activation scheme is shown to produce a similar degree of variability as previous inter-model comparisons. The model results were consistent with the idea in the literature that the CDNC in the pre-industrial run will more strongly influence the AIE than the activation scheme itself.

The authors contextualize the specific aim of this paper in an ongoing effort to quantify the AIE and uncertainty in the calculation of it. The comparison of the AIE from these three schemes as well as the heuristic for each is novel. The evaluations of model performance against measurements as well as inter-model experiments are very strong aspects of the manuscript. One limitation that the authors could more fully address is that CDNC are consistently underpredicted by MARC; although noted, the implications of this characteristic of the aerosol and activation schemes are not conveyed. Another area that could be strengthened is the introduction of and discussion about the minimum-maximum supersaturation heuristic. although the value of it as a simple scheme to introduce more variability in the activation schemes is noted, discussion about how distinct the results are for the comprehensive and heuristic Abdul-Razzak and Ghan and Morales Betancourt and Nenes schemes is lacking. I recommend this manuscript for publication in Atmospheric Chemistry and Physics with only minor changes including responses to the issues noted above and addressing the specific comments below.

*Specific Comments*

| Line | Comment |
|------|---------|
| *Line* | *Comment* |
| p. 3, l. 27-33 | The chemical constituents may need to have subscription of the numbers unless the variable names representing these compounds are being used. These also occur elsewhere (e.g., p. 5, l. 8), so please change throughout the manuscript. |
| p. 4, l. 28-29 | Please elaborate on the minimum-maximum supersaturation approach, note the description in Appendix A4, and state the motivation for implementing it. |
| p. 5, l. 8-9 | Please cite the default CESM inventory. |

p. 5, l. 20      Please cite "maximum-random overlap hypothesis" or explain it more thoroughly.

p. 6, l. 3       Please communicate whether bias may be introduced through the regridding required of the CERES dataset.

p. 10, l. 11     Please identify the particular model result when referencing a result by its qualities (and again at p. 13, l. 14).

p. 10, l. 14     "CCN" is likely intended to be "CCN."

p. 11, l. 16     "change PI and PD" is likely intended to be "change in PI and PD".

p. 11, l. 28     "couplings, and therefore different" would be better as "couplings and, therefore, different"

p. 11, l. 16     "change PI and PD" is likely intended to be "change in PI and PD".

---

## Referee Comment (RC2) · Anonymous Referee #2 · 30 Nov 2017

General comments:

Overall this paper provides a nice approach to gain insight into understanding the sensitivities of a climate model, and in particular the Aerosol Indirect Effect(AIE), to the choice of activation scheme. By using a single 'parent' model (in this instance MARC-CESM), they avoid many of the uncertainties associated with the myriad other model differences that are seen in comparisons such as the Aerocom study.

When it comes to aerosol activation schemes - and the effort and complexity that goes into refining them - my major concern is that the uncertainties in the meteorology (as well as the underlying dry aerosol population) might outweigh the detailed behaviour

of the activation scheme. While this paper does provide some information about the meteorology (e.g. LWP, cloud fraction), a little more insight into the mechanisms behind the sensitivities might be seen by exploring those aspects that have a direct impact on the activation (e.g TKE, deep convection), rather than emergent properties. Inclusion of this (see specific comment 1) would make the conclusions of the paper much more robust.

Beyond this, the paper is well written and I would expect it to be accepted for publication after considering these suggestions below.

Specific comments:

1) The benefit of a comparison that uses a common parent model (in this case MARC-CESM) is that a number of the uncertainties relating to the idiosyncratic non-linear interactions and their associated bespoke 'tunings' can be avoided. However, they cannot be removed entirely, even with the common driving model. In particular, the activation scheme is a function not only of the dry aerosol inputs, but also of the meteorology. Figure 5 aims to show some of the variation of meteorology, but it would be good to show those aspects that have a more direct impact, i.e. the updraft velocities (i.e. TKE) and the frequency of deep convection (which provides a different relationship between cloud and activation or scavenging processes). The former may explain to a certain extent the global under-prediction of CDNC, while the latter would shed light on the over-prediction in the tropics. It is apparent (but not directly commented on) that schemes that see an increase in the optical depth in the tropics (Fig 5c) also show a significant increase in the CDNC in the eastern equatorial Pacific (Fig 4). Do you see any changes in the convective activity in this region? Can you supply plots of frequency of convection or partitioning of precipitation between stratiform and convective? Ideally some plots/statistics of the above should be included, but if this information is not readily available, then some discussion relating to the role of these aspects should be included.

2) Also related to statistics on changes to the TKE; in the introduction you reference Hoose et al (2009) who show a (spurious) sensitivity to a lower bound on CDNC. You also have a lower bound in the updraft velocity of 0.2m/s. How often is this bound enforced for the different runs/schemes?

3) Figure 7b shows a significant difference between ARG and other schemes. However, there is much less difference between PCM and Nenes, i.e. schemes that are more 'sophisticated'. I think this should be highlighted, particularly where the key message in the abstract suggests an uncertainty (spread) of 0.8Wm-2. It would be interesting to know what the spread would be if you replaced ARG with other schemes such as those included in Ghan et al 2011.

Technical corrections:

P4, L30: OLS should be bold to be consistent with the other variants.

P4, L30: Could you expand what you mean 'for the supplemental heuristic'?

P5, L23: Missing close bracket

P7, L24: Could you elucidate where the enhancement by anthropogenic aerosol is captured by nenes and PCM please?

P8, L4-5: '..ensemble of aerosol-climate models,...' I presume these are the Aerocom models.

P8, L28: Do you mean Figure 7? This only shows the PD-PI. Maybe 5f?

P9, L24-25: Bracketed expression and cite are inconsistent (with spurious comma).

P10,L32: Missing close bracket.

P11,L8: I was slightly confused by this, since the correlation is positive. Perhaps rephrase as something like '...shows a correlation between pre-industrial CDNC and the indirect effect, such that $\Delta$ CRE decreases as PI CDNC increases'

P12, L3: You reference Ghan et al 2011, I think it should be Ghan et al 2013.

P14, L18: '. . .aerosol within AN adiabatically ascending. . .parcel, provideS. . .' missing words/letters suggested in caps.

P15, L13: 'form' should be from

P16, L9: 'module' should be model?

P16, L13: 'S_{math}' should be S_{max}
* * *

---

## Author Comment (AC1) · 4 Jan 2018

**Author's Response – "On the representation of aerosol activation and its influence on model-derived estimates of the aerosol indirect effect"**

We would like to thank the reviewer for their helpful suggestions and feedback on our manuscript. To streamline the comment/dialogue process, we have reproduced the comments from the reviewer and added responses inline, using blue text.

**Reviewer #1**

*General Comments*

In "On the representation of aerosol activation and its influence on model-derived estimates of the aerosol indirect effect", Rothenberg et al. integrated multiple droplet activation schemes into the Community Earth System Model (CESM) with a well-established, more-detailed aerosol module to quantify the influence of these parameterizations on the calculated aerosol indirect effect (AIE). The aerosol module tracks the evolution of number and mass for lognormally-distributed sulfate (three modes), black carbon, organic carbon, mixed modes, dust, and sea salt. The aerosol and droplets are coupled with radiative transfer and cloud microphysics, which are representative of stratiform clouds. The authors integrate three activation schemes as well as derivations of each scheme that employ a heuristic for more quickly calculating activation based on the dominant mode. They evaluate the predicted cloud droplet number concentration (CDNC) against in situ and satellite-based observations and demonstrate that these observations are insufficient for identifying a singular scheme as successful. Then, they comprehensively and clearly assess the AIE, in which they find a doubling of the strongest compared with the weakest result, and the parameters influencing AIE. Comparing the dependence of the change in cloud radiative effect (CRE) on other parameters and the degree of spread in the AIE in these simulations to the range of AeroCom and IPCC, the choice of activation scheme is shown to produce a similar degree of variability as previous inter-model comparisons. The model results were consistent with the idea in the literature that the CDNC in the pre-industrial run will more strongly influence the AIE than the activation scheme itself.

The authors contextualize the specific aim of this paper in an ongoing effort to quantify the AIE and uncertainty in the calculation of it. The comparison of the AIE from these three schemes as well as the heuristic for each is novel. The evaluations of model performance against measurements as well as inter-model experiments are very strong aspects of the manuscript. One limitation that the authors could more fully address is that CDNC are consistently underpredicted by MARC; although noted, the implications of this characteristic of the aerosol and activation schemes are not conveyed…

We acknowledge this limitation and have added a paragraph in the Discussion and Conclusions section of the manuscript contextualizing this. We defer a more complete discussion of this topic to a follow-up work comparing a rigorous comparsion of MARC to CESM/MAM3 and CESM/MAM7, which is currently in the final stages of preparation before submission to for peer review.

… Another area that could be strengthened is the introduction of and discussion about the minimum-maximum supersaturation heuristic. although the value of it as a simple scheme to introduce more variability in the activation schemes is noted, discussion about how distinct the results are for the comprehensive and heuristic Abdul-Razzak and Ghan and Morales Betancourt and Nenes schemes is lacking…

In response to one of the specific comments below, we added some additional details on the motivation and mechanics of the minimum-maximum supersaturation heuristic. We have also added a bit more discussion on this topic in the manuscript (towards the end of Section 3.3 and in the Discussion and Conclusions), particularly focused on a result best illustrated by Figures 6b and 8, which show that for both the *ARG* and *nenes* schemes, the *_min_smax* heuristic tends to increase the change in CDNC from PI to PD, but the associated change in SW radiative forcing goes in opposite directions.

… I recommend this manuscript for publication in Atmospheric Chemistry and Physics with only minor changes including responses to the issues noted above and addressing the specific comments below.

*Specific Comments*

- P. 3, L. 27-33: The chemical constituents may need to have subscription of the numbers unless the variable names representing these compounds are being used. These also occur elsewhere (e.g., p. 5, l. 8), so please change throughout the manuscript.

  We appreciate catching these errors; these are a mistake from our conversion of the LaTeX source of the manuscript to comply with ACP's technical restrictions. The entire manuscript was reviewed to find and correct this common mistake, as well as related ones involving incorrect superscripts with units.

- P. 4, L. 28-29: Please elaborate on the minimum-maximum supersaturation approach, note the description in Appendix A4, and state the motivation for implementing it.

  We have modified the paragraph to include an explicit reference to the appendix and additional discussion of the motivation and mechanics, based on results from Rothenberg and Wang (2017).

- P. 5, L. 8-9: Please cite the default CESM inventory.

We've added a reference to (Lamarque et al., 2010).

- P. 5, L. 20: Please cite "maximum-random overlap hypothesis" or explain it more thoroughly.

  We've added a short explanation of this hypothesis, as well as a citation to literature which more completely explains how the assumption is formulated (Morcrette, 1991).

- P. 6, L. 3: Please communicate whether bias may be introduced through the regridding required of the CERES dataset.

  We have added an acknowledgement that re-gridding in this case, which involves *downsampling* to the CESM/MARC grid, will tend to suppress or average out regional variability, which will introduce some biases in the CERES data.

- P. 10, L. 11: Please identify the particular model result when referencing a result by its qualities (and again at p. 13, l. 14).

  We've re-phrased the sentences to identify specific models/simulations in both of these cases.

- P. 10, L. 14: "CCN" is likely intended to be "CCN."

  Fixed.

- P. 11, L. 16: "change PI and PD" is likely intended to be "change in PI and PD".

  Fixed.

- P. 11, L. 28: "couplings, and therefore different" would be better as "couplings and, therefore, different"

  Adopted the recommended punctuation.

**References**

Lamarque, J.-F., Bond, T. C., Eyring, V., Granier, C., Heil, A., Klimont, Z., Lee, D., Liousse, C., Mieville, A., Owen, B., Schultz, M. G., Shindell, D., Smith, S. J., Stehfest, E., Van Aardenne, J., Cooper, O. R., Kainuma, M., Mahowald, N., McConnell, J. R., Naik, V., Riahi, K. and van Vuuren, D. P.: Historical (1850–2000) gridded anthropogenic and biomass burning emissions of reactive gases and aerosols: methodology and application, Atmos. Chem. Phys., 10(15), 7017–7039, doi:10.5194/acp-10-7017-2010, 2010.

Rothenberg, D. and Wang, C.: An aerosol activation metamodel of v1.2.0 of the pyrcel cloud parcel model: Development and offline assessment for use in an aerosol-climate model, Geosci. Model Dev. Discuss., 1–35, doi:10.5194/gmd-2016-228, 2016.

---

## Author Comment (AC2) · 4 Jan 2018

**Author's Response – "On the representation of aerosol activation and its influence on model-derived estimates of the aerosol indirect effect"**

We would like to thank the reviewer for their helpful suggestions and feedback on our manuscript. To streamline the comment/dialogue process, we have reproduced the comments from the reviewer and added responses inline, using blue text. We have addressed a list of "Specific Comments" directly, in similar fashion, along with recommended technical corrections

**Reviewer #2**

*General Comments*

Overall this paper provides a nice approach to gain insight into understanding the sensitivities of a climate model, and in particular the Aerosol Indirect Effect(AIE), to the choice of activation scheme. By using a single 'parent' model (in this instance MARC- CESM), they avoid many of the uncertainties associated with the myriad other model differences that are seen in comparisons such as the Aerocom study.

When it comes to aerosol activation schemes - and the effort and complexity that goes into refining them - my major concern is that the uncertainties in the meteorology (as well as the underlying dry aerosol population) might outweigh the detailed behavior of the activation scheme. While this paper does provide some information about the meteorology (e.g. LWP, cloud fraction), a little more insight into the mechanisms behind the sensitivities might be seen by exploring those aspects that have a direct impact on the activation (e.g TKE, deep convection), rather than emergent properties. Inclusion of this (see specific comment 1) would make the conclusions of the paper much more robust.

We acknowledge this key point as raised by the reviewer, and address it in more detail as noted in Specific Comment (1).

Beyond this, the paper is well written and I would expect it to be accepted for publication after considering these suggestions below.

*Specific comments*

1) The benefit of a comparison that uses a common parent model (in this case MARC- CESM) is that a number of the uncertainties relating to the idiosyncratic non-linear interactions and their associated bespoke 'tunings' can be avoided. However, they cannot be removed entirely, even with the common driving model. In particular, the activation scheme is a function not only of the dry aerosol inputs, but also of the meteorology. Figure 5 aims to show some of the variation of meteorology, but it would be good to show those aspects that have a more direct impact, i.e.

the updraft velocities (i.e. TKE) and the frequency of deep convection (which provides a different relationship between cloud and activation or scavenging processes). The former may explain to a certain extent the global under-prediction of CDNC, while the latter would shed light on the over-prediction in the tropics. It is apparent (but not directly commented on) that schemes that see an increase in the optical depth in the tropics (Fig 5c) also show a significant increase in the CDNC in the eastern equatorial Pacific (Fig 4). Do you see any changes in the convective activity in this region? Can you supply plots of frequency of convection or partitioning of precipitation between stratiform and convective? Ideally some plots/statistics of the above should be included, but if this information is not readily available, then some discussion relating to the role of these aspects should be included.

Unfortunately, we don't have the diagnostics for frequency of convection from these runs (we prioritized high-frequency aerosol and microphysical diagnostics). We do note that we've made an effort in past work (Rothenberg and Wang, 2016) to rigorously quantify the influence of meteorology (through vertical updraft velocity) on CDNC using the activation schemes included here, through its interactions with the dry aerosol input parameters, but we agree that in the context of online climate model simulations, the uncertainties arising from interactions with meteorology are a key part of the story here.

To comment specifically on Figure 4 – the "hot spots" of CDNC in the equatorial Eastern Pacific here are not necessarily increases in CDNC; rather, they are areas where the model-derived CDNC is much greater than that estimated from MODIS (Bennartz, 2007). Complicating the interpretation here, this is an area where the MODIS-derived inventory likely under-predicts CDNC itself (and shows very low CDNC values in Figure 4) because it uses an idealized cloud model and set of assumptions in its inference. The large "relative error" in CDNC here is partially a numerical artifact given the small values present in the observational dataset.

To zero in a bit more on potential changes in convective activity in this region due to the choice of activation scheme, we plot the present-day, annual average convective precipitation rates for the **arg_comp** case, followed by the absolute difference between this field and those associated with each of the other 6 activation schemes (similar to Figure 7b):

[Figure]

Figure 1) Annual average convective precipitation rate in the **arg_comp** case

[Figure]

Figure 2) Difference in annual average convective precipitation rate between **arg_comp** case and cases indicated on each panel

In the Eastern equatorial Pacific, the difference in convective precipitation activity between each scheme is very small. We clarify some of these results in the discussion of Figure 5c in the manuscript. Regarding the potential distribution of TKE/sub-grid scale vertical velocity, we defer this point to the following comment.

2) Also related to statistics on changes to the TKE; in the introduction you reference Hoose et al (2009) who show a (spurious) sensitivity to a lower bound on CDNC. You also have a lower bound in the updraft velocity of 0.2m/s. How often is this bound enforced for the different runs/schemes?

For a more complete discussion of updraft vertical velocity, TKE, and its relation to activation within MARC, we refer the reviewer to Section 2.1 of Rothenberg and Wang (2016). To directly answer the reviewer's question, we reproduce here Figure 1 of that work, which shows PDFs of updraft vertical velocity for near-surface grid cells (below 700 mb), broken down into land (red) and ocean (black) regimes, for a reference MARC simulation using the *arg_comp* activation scheme:

[Figure]

Figure 3 - Distributions of model-predicted instantaneous subgrid-scale vertical velocity for `near-surface (below 700mb) grid cells` broken down by land (red) and ocean (black) regimes.

The lower bound of 0.2 m/s is frequently invoked over both land and ocean regimes, although it is more frequent over land than ocean. We note that West et al. (2014) showed that TKE-based parameterizations do a good job of reproducing the spatio-temporal variability of updraft velocity, but tend to produce an unrealistically high frequency of any minimum threshold imposed on that velocity.

The results we have previously reported looking at offline activation diagnostics across the complete aerosol-meteorology parameter space sampled by MARC suggest that MARC's under-prediction of CDNC arises from a systematic bias towards fewer, smaller aerosol particles. Put another way, we simulate fewer CCN and as a result, fewer CDNC. A follow-up work rigorously comparing MARC to the CESM/MAM3 and CESM/MAM7 which is in preparation for submission for peer review looks at this issue in more detail.

3) Figure 7b shows a significant difference between ARG and other schemes. However, there is much less difference between PCM and Nenes, i.e. schemes that are more 'sophisticated'. I think this should be highlighted, particularly where the key message in the abstract suggests an uncertainty (spread) of 0.8Wm-2. It would be interesting to know what the spread would be if you replaced ARG with other schemes such as those included in Ghan et al 2011.

We have made some changes to the Discussion and Conclusions to emphasize this point, although we are careful not to suggest that 'sophisticated' here is equivalent to 'better', especially with regards to producing a more reliable estimate of the indirect effect. We agree

that it would be useful and very interesting to supplement the analyses presented here with additional simulations incorporating the full bevy of activation schemes documented by Ghan et al (2011); however, we faced some difficulties in obtaining implementations and/or independently reproducing some of the results previously reported by those schemes, hence the abbreviated list we studied here and in the works preceding this manuscript.

We hope to conduct this exercise in the future with additional activation schemes and aerosol-climate models.

*Technical corrections:*

- P4, L30: OLS should be bold to be consistent with the other variants.

  Tweaked the phrasing in the sentence to clarify that this refers to **pcm_ols4**.

- P4, L30: Could you expand what you mean 'for the supplemental heuristic'?

  Re-worded the sentence to explicitly mention that this is the minimum-Smax heuristic.

- P5, L23: Missing close bracket

  Fixed typo.

- P7, L24: Could you elucidate where the enhancement by anthropogenic aerosol is captured by nenes and PCM please?

  In this comment, we wish to emphasize two locations: the coast of eastern Asia, and eastern United States. We have added a clarifying remark in the manuscript.

- P8, L4-5: '..ensemble of aerosol-climate models,. . .' I presume these are the Aerocom models.

  Yes; we have added a remark to clarify this.

- P8, L28: Do you mean Figure 7? This only shows the PD-PI. Maybe 5f?

  We mean Figure 7**b**, which we use to compare the spread in SW CRE for the present-day between each scheme versus the change in SW CRE between PI and PD (show in in 7a). We have updated the reference as such.

- P9, L24-25: Bracketed expression and cite are inconsistent (with spurious comma).
-
  We fixed an error in the LaTeX parenthetical citation macro which lead to this typo.

- P10,L32: Missing close bracket.

  Fixed.

- P11,L8: I was slightly confused by this, since the correlation is positive. Perhaps rephrase as something like '. . .shows a correlation between pre-industrial CDNC and the indirect effect, such that Δ CRE decreases as PI CDNC increases'

  Fixed using a phrasing similar to the proposed modification.

- P12, L3: You reference Ghan et al 2011, I think it should be Ghan et al 2013.

  We do mean Ghan et al (2011) here; the reduction we're referring to is summarized in Section 5, P. 19 (second paragraph), where it is reported that the ARG scheme yields SW CRE of -1.76 W/m$^2$ but the Fountoukis and Nenes (2005) scheme produces a smaller one of -1.60 W/m$^2$

- P14, L18: '. . .aerosol within AN adiabatically ascending. . .parcel, provideS. . .' missing words/letters suggested in caps.

  We've included the suggested additional words/letters.

- P15, L13: 'form' should be from

  Fixed.

- P16, L9: 'module' should be model?

  Yes; fixed.

- P16, L13: 'S_{math}' should be S_{max}

  Fixed.

**References**

Bennartz, R.: Global assessment of marine boundary layer cloud droplet number concentration from satellite, J. Geophys. Res., 112(D2), D02201, doi:10.1029/2006JD007547, 2007.

Lamarque, J.-F., Bond, T. C., Eyring, V., Granier, C., Heil, A., Klimont, Z., Lee, D., Liousse, C.,

Mieville, A., Owen, B., Schultz, M. G., Shindell, D., Smith, S. J., Stehfest, E., Van Aardenne, J., Cooper, O. R., Kainuma, M., Mahowald, N., McConnell, J. R., Naik, V., Riahi, K. and van Vuuren, D. P.: Historical (1850–2000) gridded anthropogenic and biomass burning emissions of reactive gases and aerosols: methodology and application, Atmos. Chem. Phys., 10(15), 7017–7039, doi:10.5194/acp-10-7017-2010, 2010.

Morcrette, J.-J.: Radiation and cloud radiative properties in the European Centre for Medium Range Weather Forecasts forecasting system, J. Geophys. Res., 96(D5), 9121, doi:10.1029/89JD01597, 1991.

Rothenberg, D. and Wang, C.: An aerosol activation metamodel of v1.2.0 of the pyrcel cloud parcel model: Development and offline assessment for use in an aerosol-climate model, Geosci. Model Dev. Discuss., 1–35, doi:10.5194/gmd-2016-228, 2016.

West, R. E. L., Stier, P., Jones, A., Johnson, C. E., Mann, G. W., Bellouin, N., Partridge, D. G. and Kipling, Z.: The importance of vertical velocity variability for estimates of the indirect aerosol effects, Atmos. Chem. Phys., 14(12), 6369–6393, doi:10.5194/acp-14-6369-2014, 2014.

---

## Author Response (AR2)

**Author's Response – "On the representation of aerosol activation and its influence on model-derived estimates of the aerosol indirect effect"**

We again appreciate the thoughtful and thorough review and comments from both the reviewer and the editor; they produced very fruitful discussion among the authors about the results reported in the manuscript, as well as many ideas for future investigation.

After carefully considering the comments, we reviewed once more the model output archived from these simulations to assess what additional diagnostics we might be able to perform to explore and quantify the potential changes in convective activity. Unfortunately, we are somewhat limited by our original modeling/experimental design; we prioritized high-frequency, 3D cloud and aerosol microphysical outputs in this work, and couldn't easily identify suitable diagnostics in our dataset beyond the convective precipitation data. We will certainly keep this in mind when conducting future modeling studies concerning the aerosol indirect effect.

Therefore, we've opted to closely follow the reviewer's suggestion to include a subsection in Section 3 to fully discuss the potential confounding role of changes in meteorology on isolating the role of the aerosol activation schemes. We include both Figure 1 and Figure 2 from the original response (Figure 12 and 13 in the revised manuscript) as a focus of the discussion in this section. The final sub-section of Section 3, "Summarizing the influence of aerosol activation", also includes reference to this discussion. Pursuant with these changes, we've also adopted the reviewer's recommendation for contextualizing the role of meteorology in the abstract, and added a short discussion in the concluding section of the manuscript to discuss future work which might be able to help better separate the meteorological feedback from changes arising from the different activation schemes.

We regret that we are not able to conduct a more thorough analysis at this time to explore the important and interesting issue of meteorological feedbacks arising from the subtle changes in activation schemes in our simulations. However, we believe that the included section and brief analysis sufficiently raises attention to the potential role of the models' meteorological response in shaping our results.

*Technical corrections:*

- Page 9, line 4: '...play online a minor role...' should be '...play only a minor role...'?

  Yes; this typo has been fixed.

[revised manuscript text omitted]